# Evolution of brilliant iridescent feather nanostructures

**Klara Katarina Nordén**[1]*, **Chad M Eliason**[2], **Mary Caswell Stoddard**[1]

[1]Department of Ecology and Evolutionary Biology, Princeton University, Princeton, United States; [2]Grainger Bioinformatics Center, Field Museum of Natural History, Chicago, United States

**Abstract** The brilliant iridescent plumage of birds creates some of the most stunning color displays known in the natural world. Iridescent plumage colors are produced by nanostructures in feathers and have evolved in diverse birds. The building blocks of these structures—melanosomes (melanin-filled organelles)—come in a variety of forms, yet how these different forms contribute to color production across birds remains unclear. Here, we leverage evolutionary analyses, optical simulations, and reflectance spectrophotometry to uncover general principles that govern the production of brilliant iridescence. We find that a key feature that unites all melanosome forms in brilliant iridescent structures is thin melanin layers. Birds have achieved this in multiple ways: by decreasing the size of the melanosome directly, by hollowing out the interior, or by flattening the melanosome into a platelet. The evolution of thin melanin layers unlocks color-producing possibilities, more than doubling the range of colors that can be produced with a thick melanin layer and simultaneously increasing brightness. We discuss the implications of these findings for the evolution of iridescent structures in birds and propose two evolutionary paths to brilliant iridescence.

## Editor's evaluation

Nordén et al., examine feather iridescent color diversity across bird species. Their findings show how key modifications in feather melanosomes, pivotal nanophotonic structures, underlie the brilliant colors of iridescent feathers, broadening feather color range approximately twofold. In a next step, the authors evaluate the function of feather melanosomes by performing optical modeling of nanostructure diversity, evaluating up to 4500 distinct nanostructure combinations, which are then contrasted with the observed (color) spectral data from 111 plumage regions across 80 (diverse) bird species. This meticulous integration of diverse methods across a comprehensive dataset will not only inform biologists studying structural color biodiversity, but it may also inspire engineers designing nanophotonic systems.

*For correspondence: knorden@princeton.edu

Competing interest: The authors declare that no competing interests exist.

## Introduction

Many animal colors—and indeed some plant, algae, and possibly fungus colors (**Brodie et al., 2021**)—are structural, produced by the interaction of light with micro- and nano-scale structures (reviewed in **Kinoshita et al., 2008**). In birds, structural colors greatly expand—relative to pigment-based mechanisms—the range of colors birds can produce with their feathers (**Stoddard and Prum, 2011**; **Maia et al., 2013b**). Some structural colors are iridescent: the perceived hue changes with viewing or lighting angle. Iridescent coloration features prominently in the dynamic courtship displays of many bird species, including birds-of-paradise (Paradisaeidae), hummingbirds (Trochilidae), and pheasants (Phasianidae) (**Greenewalt et al., 1960**; **Stavenga et al., 2015**; **Zi et al., 2003**). These dazzling displays showcase the kind of bright and saturated iridescent colors that have previously been qualitatively

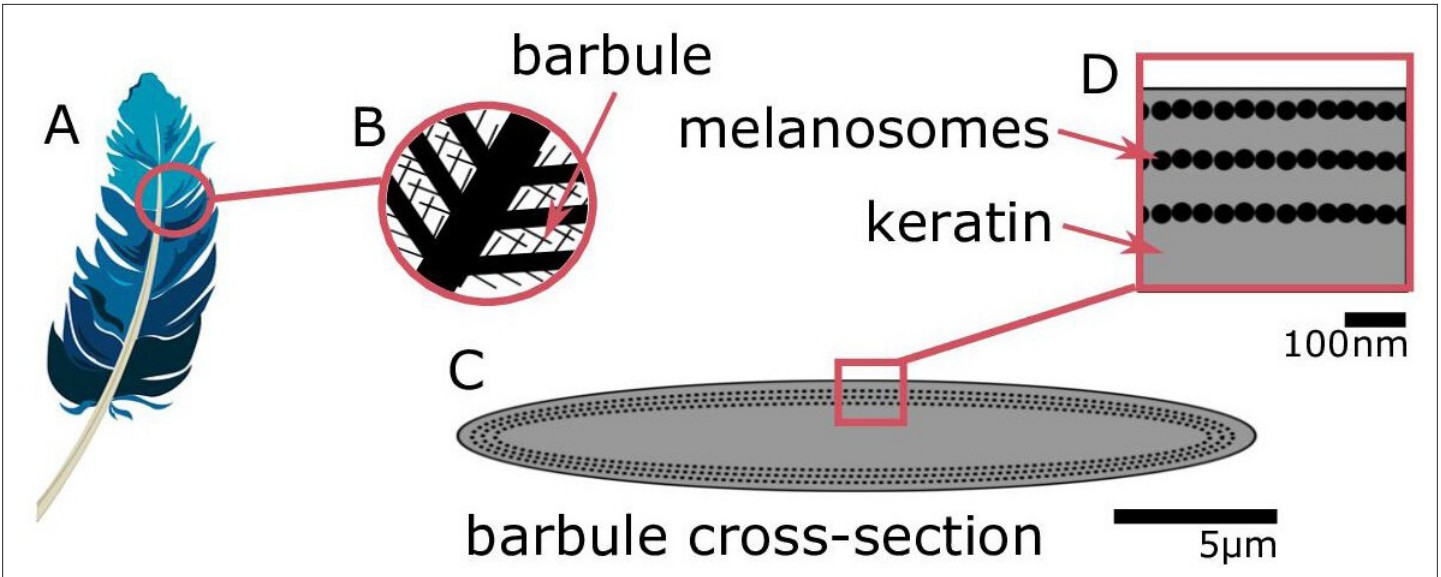

**Figure 1.** Iridescent plumage is produced by nanostructures in the feather barbules. A vaned feather (**A**) consists of branching structures where the barbules (**B**) are the interlocking filaments. A cross-section of a barbule from an iridescent feather (**C**) reveals the intricate nanostructure responsible for the color, consisting of layers of melanosomes in keratin (**D**).

categorized as 'luxurious' (*Auber, 2008*) or 'brilliant' (*Durrer, 1977*), in contrast to the more muted 'faint' (*Auber, 2008*) or 'weak' (*Durrer, 1977*) iridescent colors of, for example, a brown-headed cowbird (*Molothrus ater*). Following these authors, we use the terms 'brilliant' and 'weak' to describe this difference in color appearance, where brilliant iridescence describes colors of high saturation and brightness and weak iridescence describes colors of low saturation and brightness.

Typically, brilliant iridescence is associated with more complex feather nanostructures, relative to weak iridescence. All iridescent feather coloration is produced by nanostructures in the feather barbules consisting of melanin-filled organelles (melanosomes) and keratin (*Figure 1*), but brilliant iridescent coloration arises from light interference by photonic crystal-like structures (henceforth photonic crystals), while weak iridescent coloration is produced by structures with a single layer of melanosomes (*Durrer, 1977*). A photonic crystal is defined by having periodic changes in refractive index (*Joannopoulos et al., 2008*); in feather barbules, this is created by periodic arrangements of melanosomes in keratin. By adding more reflection interfaces, a photonic crystal greatly amplifies color saturation and brightness compared to a single-layered structure, the latter of which typically functions as a simple thin film (*Kinoshita et al., 2008*). Thus, brilliant iridescence describes bright, highly saturated colors arising from melanosomes arranged in a photonic crystal.

In iridescent feathers, it is not just the arrangement of melanosomes that can vary: the melanosomes also come in a variety of different shapes. *Durrer, 1977* classified melanosomes into five main types: (1) thick solid rods (S-type, *Figure 2A*); (2) thin solid rods (St-type, *Figure 2B*); (3) hollow rods (with an air-filled interior, R-type, *Figure 2C*); (4) platelets (P-type, *Figure 2D*); and (5) hollow platelets (K-type, *Figure 2E*). All five melanosome types occur in single-layered structures producing weak iridescence. Four of these types—all but thick solid rods—occur in photonic crystals producing brilliant iridescence. This diversity is extraordinary given that the shape of melanosomes in other melanized vertebrate tissues, including black and gray feathers, is typically a solid rod (*D'Alba and Shawkey, 2019*). The thick solid rods found in weakly iridescent feathers resemble the melanosomes found in plain black feathers (*Durrer, 1977*) and are likely ancestral to the four more modified, derived melanosome shapes (*Shawkey et al., 2006*; *Maia et al., 2012*). Because the derived melanosome shapes (but not the ancestral thick solid rods) are arranged as photonic crystals, these two innovations together—novel shapes and photonic crystal structure—may have been critical for the evolution of brilliant iridescence. Supporting this idea, *Maia et al., 2013b* showed that the evolution of

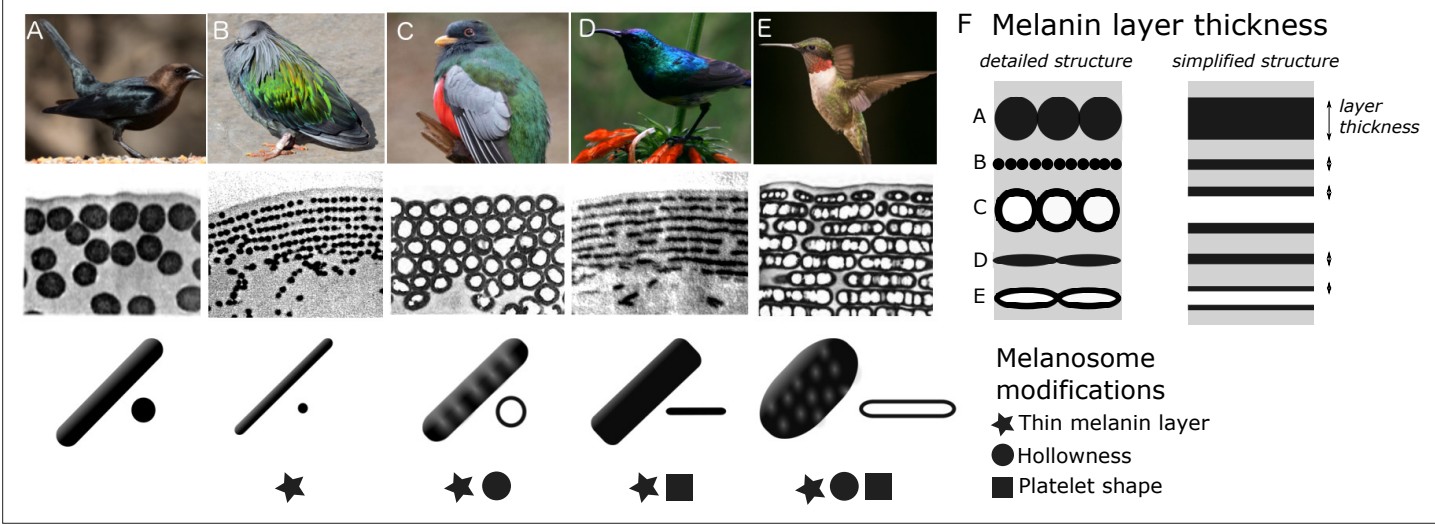

**Figure 2.** Iridescent feather nanostructures are diverse. Structures can vary both in melanosome type and melanosome organization. There are five main types of melanosomes (shown as schematics in bottom row, each viewed from the side and in cross-section (**A–E**)) and two main types of structural organization (shown by microscope images of barbule cross-sections, middle row: single-layered (**A**) and photonic crystal (**B–E**)). A single-layered structure with thick solid rods (**A**) gives rise to the dark, black-blue iridescence of a brown-headed cowbird (*Molothrus ater*). This type of structure generally gives rise to 'weak' iridescent colors, with low color saturation and brightness. Photonic crystals (**B–E**) with multiple layers of melanosomes generally give rise to 'brilliant' iridescent colors, with high saturation and brightness. Thin solid rods (**B**) in a multilayer configuration (also called a one-dimensional photonic crystal) produce the iridescent colors of the Nicobar pigeon (*Caloenas nicobarica*). In the elegant trogon (*Trogon elegans*), the iridescent green color is produced by hexagonally packed hollow rods (**C**). Sunbird (here the variable sunbird, *Cinnyris venustus*) barbules contain melanosomes stacked in multilayers, with solid platelet-shaped melanosomes serving as the building blocks (**D**). The fifth melanosome type is a hollow platelet (**E**), which forms multilayer configurations in many hummingbird species (here a ruby-throated hummingbird, *Archilochus colubris*). The five types of melanosomes are characterized by different combinations of three key modifications: thin melanin layers, hollowness, and platelet shape, which are indicated as symbols under each melanosome type. Thin melanin layers are present in four melanosome types, but they are achieved in different ways, as is shown by the schematic in (**F**). A simplified diagram of each melanosome type (**F**, right) shows how solid forms translate to a single melanin layer, while hollow forms create two thinner melanin layers intersected by an air layer. All photographs (top row) are under a Public Domain License (details in *Supplementary file 1n*).

hollow and/or platelet-shaped melanosomes in African starlings (Sturnidae) was associated with great expansions in color diversity and increases in brilliance. Moreover, *Eliason et al., 2013* used optical modeling and plumage color measurements of the violet-backed starling (*Cinnyricinclus leucogaster*) and wild turkey (*Meleagris gallopavo*) to show that hollow rods increase the brightness of iridescent colors compared to structures with solid rods.

While previous studies focusing on nanostructural evolution and color-producing mechanisms in a variety of avian groups (*Eliason et al., 2020*; *Eliason et al., 2015*; *Eliason et al., 2013*; *Eliason and Shawkey, 2012*; *Gammie, 2013*; *Gruson et al., 2019*; *Maia et al., 2013b*; *Quintero and Espinosa de los Monteros, 2011*) have given us valuable insights into the evolution and optics of iridescent structures, they have focused on specific species, small clades, or a particular melanosome type. Thus, they have not uncovered the broader, general principles governing the evolution of brilliant iridescent plumage, and several key questions remain unanswered.

Why have bird species with brilliant iridescence evolved not one but four different melanosome types? How are these melanosome types phylogenetically distributed? Are particular melanosome types associated with different plumage colors? Since Durrer's initial work in 1977, there has been no broad-scale evolutionary analysis of the melanosomes in iridescent feathers, and no study has compared the optical effects of all five of Durrer's melanosome types. To find general principles underlying differences in color production, we identify key modifications that, based on optical theory, are likely to be important. This enables us to compare the five melanosome types rigorously, since each type can have several modifications. For example, a hollow platelet (*Figure 2E*) has both an air-filled interior and a flattened shape, both of which might influence feather color—perhaps in different ways.

Therefore, a simple comparison of the melanosome types cannot reveal which modifications affect color production or pinpoint their precise optical effects.

In this study, we search for general design principles underlying the production of brilliant iridescent coloration. First, we identify three key modifications of melanosomes in brilliant iridescent structures: thin melanin layers, hollowness, and platelet shape (*Figure 2*). Second, we create a feather iridescence database using published descriptions of iridescent feather structures. Using the database, we explore the evolutionary history of the three key modifications of brilliant iridescent structures. Third, we use optical modeling to simulate colors that could be produced with each melanosome type; we estimate light reflectance from 4500 different structures using parameter ranges derived from the database. Finally, we analyze spectral data from 111 plumage regions across 80 diverse bird species with known nanostructures to test the predictions of our optical model.

## Results

### Identifying key melanosome modifications

The size, composition, and shape of materials that form the periodic layers in a photonic crystal can all contribute to its reflectance properties (*Joannopoulos et al., 2008*). In iridescent structural feather colors, the layers are formed by melanosomes, and we can identify three melanosome modifications that likely have important optical effects. We define these modifications relative to the thick solid rods found in weakly iridescent feathers, since we presume these to be unmodified or minimally modified from melanosomes found in other non-iridescent melanized tissues, which they closely resemble (*Durrer, 1977*). For a more detailed analysis of thick solid rods in weakly iridescent versus black feathers, see the next section (***Evolution of modified melanosomes in iridescent structures***). The three modifications are: thin melanin layers (size of layers), an air-filled interior (layer material composition), and platelet shape (shape of layers). 'Thin' here refers to something thinner than the ancestral thick solid rods. A 'melanin layer' refers to a single layer in the optical structure. For solid rods and platelets, a layer's thickness is simply the rod or platelet diameter, but for hollow rods and platelets, it is the thickness of a single melanin wall (*Figure 2F*). Each of Durrer's five melanosome types can be described in terms of the absence/presence of one or several modifications (*Figure 2*).

What are the potential optical advantages of melanosomes with these features? First let us consider thin melanin layers. Thin melanin layers may tune the structure so that it reflects optimally in the bird-visible spectrum. This possibility was raised by *Durrer, 1977*, who noted that structures producing brilliant iridescent colors tended to have thin (melanin) layers. However, Durrer's work is only available in German, and this idea has remained largely overlooked. We refine and extend Durrer's idea here using established optical theory, specifically multilayer optics (reviewed in *Kinoshita, 2008*; *Kinoshita et al., 2008*). To produce first-order interference peaks, which will result in brighter colors than higher-order interference peaks, the optical thickness (thickness×refractive index) of each repeating unit in a one-dimensional photonic crystal (also often termed multilayer, *Figure 2B and D–E*) should approximate half a wavelength $\left(\frac{\lambda}{2}\right)$ (*Durrer, 1977*; *Kinoshita et al., 2008*; *Land, 1972*). The repeating unit in an iridescent feather nanostructure consists of one layer of melanosomes and one layer of keratin, and we can therefore express this as $\left(t_{mel} \times n_{mel}\right) + \left(t_k \times n_k\right) = \frac{\lambda}{2}$, where $t_{mel}$ is the thickness of the melanin layer, $t_k$ is the thickness of the keratin layer, $n_{mel}$ is the refractive index of the melanin layer, and $n_k$ is the refractive index of the keratin layer. Among the configurations that satisfy this condition, maximum reflection is achieved when both layers have equal optical thickness, which can be expressed as $\left(t_{mel} \times n_{mel}\right) = \left(t_k \times n_k\right) = \frac{\lambda}{4}$ (*Kinoshita et al., 2008*; *Land, 1972*). From this, we can express the range within which we would expect melanin optical layer thickness to fall as $\left(t_{mel} \times n_{mel}\right) < \frac{\lambda}{2}$, with maximum reflectance at $\left(t_{mel} \times n_{mel}\right) = \frac{\lambda}{4}$. If we assume that the structure should reinforce wavelengths within the bird-visible spectrum (300–700 nm), we can calculate the range we should expect for melanin layer thickness, using 300 nm and 700 nm as endpoints. Here, we use the refractive indices $n_{mel} = 2$ for 300 nm and $n_{mel} = 1.7$ for 700 nm, following *Stavenga et al., 2015*. This gives us a maximum melanin layer thickness ranging from <75 nm (maximum thickness for reinforcing ultraviolet wavelengths) to <206 nm (maximum thickness for reinforcing red wavelengths), with maximum reflectance (where $\left(t_{mel} \times r_{mel}\right) = \frac{\lambda}{4}$) at layer thicknesses of 37.5 nm and 103 nm, respectively. Note that the maximum values of 206 nm

and 75 nm represent situations where the optical thicknesses of melanin layers alone equal $\frac{\lambda}{2}$, and thus keratin layers must be zero. Such a structure would not function as a photonic crystal, since it consists of a single thick layer of melanin. Thus, for iridescent structures producing first-order interference peaks, we expect melanin layer thickness to be below 206 nm. Moreover, we expect a lower limit at 37.5 nm, since melanin layer thickness is unlikely to have evolved below the thickness required for maximum reflectance at ultraviolet wavelengths. This gives us an expected range of 37.5–206 nm. The typical diameter of melanosomes found in vertebrates is ~300 nm (*Li et al., 2014*), exceeding this range.

Now let us consider why melanosomes with hollow interiors might be advantageous. A hollow interior could increase reflectance by creating a sharper contrast in refractive index in the structure (*Durrer, 1977*; *Eliason et al., 2013*; *Kinoshita, 2008*; *Land, 1972*; *Stavenga et al., 2018*), making a color brighter. This is because the refractive index of air (n=1) is lower than that of keratin (n=1.56).

To estimate the expected thickness of hollow interiors (air pockets), we can extend the argument for expected thickness of the melanin layer. If air pockets conform to the expected size range, this would suggest that they are tuned together with melanin layers to produce brilliant iridescence. Analogous to describing melanin rods as a melanin layer (*Figure 2F*), we can think of air pockets as an air layer. Since the equations above define reflection for a structure with only two materials (of high and low refractive index, respectively), we must assume that air layers have the same optical thickness as the keratin layers. Thus, both the keratin and air layers can be described by a single term, since $(t_k \times n_k) = (t_a \times n_a)$, where $a$ denotes air and $n_a = 1$. In this situation, the air layer should have a thickness <350 nm to produce first-order interference in the bird-visible spectrum—and a thickness of 75–175 nm to meet the condition for maximal reflectance. Thus, the expected range is 75–350 nm. However, we note that a one-dimensional photonic crystal with three materials could have varying optical thickness for all three types of layers (where $(t_k \times n_k) \neq (t_a \times n_a)$). The optimal configuration of such a system is much harder to derive, making it difficult to generate specific predictions for this case.

Finally, let us explore why platelet-shaped melanosomes might be beneficial. Platelet-shaped melanosomes have been hypothesized to increase reflection by creating smooth, mirror-like reflection surfaces (*Durrer, 1977*; *Land, 1972*). Moreover, the thin platelet shape might allow for more layers to be packed within a photonic crystal, which would increase total reflection (*Maia et al., 2013b*).

Which of the four derived melanosome types in brilliant iridescent feathers possess these modifications? Hollowness and platelet shape are each present in two types, but thin melanin layers are likely shared by all four derived melanosome types (*Figure 2*). *Durrer, 1977*, noted the prevalence of thin melanin layers but never analyzed them formally. Nonetheless, this potential convergence on thin melanin layers hints at the intriguing possibility that the four derived melanosome types present diverse paths to the same end: achieving optimal melanin layer thickness. A hollow interior or a platelet shape may simply be different mechanisms for reducing melanin layer thickness. This would also explain why thick solid rods are typically only found in single-layered structures. Single-layered structures typically function as thin films, where the thickness of the overlying keratin cortex determines the interference colors (*Doucet et al., 2006*; *Lee et al., 2012*; *Maia et al., 2009*; *Yin et al., 2006*). The layer of melanosomes only functions to delimit the keratin layer, so the thickness of the melanin layer itself is largely irrelevant. Thus, there would be no selection pressure to decrease melanin layer thickness in single-layered structures, and we would expect the ancestral condition (thick solid rods) to remain.

We suggest that the diverse melanosome types found in brilliant iridescent structures evolved to generate thin melanin layers in different ways. This possibility has not been investigated previously, probably because melanosome types are generally analyzed on the basis of their overall morphology rather than—as we have proposed here—on the basis of specific optical modifications.

## Evolution of modified melanosomes in iridescent structures

We surveyed the literature for all published descriptions of iridescent feather structures—including weak and brilliant iridescent colors—in order to build a species-level database (henceforth the feather iridescence database) of key structural parameters (Figure 8). These parameters included melanosome type (solid rod, hollow rod, solid platelet, and hollow platelet), melanin layer thickness, details about the structure (single-layered or photonic crystal), and size of the internal air pockets. We found

that iridescent feather nanostructures have been described in 306 bird species representing 15 different orders and 35 families. The feather iridescence database, which includes a complete list of the references we consulted, is available to download from the Dryad Digital Repository (doi.10.5061/dryad.4j0zpc8bq).

Descriptions of iridescent feather structures are taxonomically biased, with some groups well represented (>20 species represented in the database: Sturnidae, Trochilidae, Phasianidae, Trogonidae, and Anatidae) but most groups sparsely sampled (<5 species represented in the database) or absent despite some species possessing iridescent plumage (e.g., Picidae). Even in well-sampled groups (e.g., Trochilidae), the feather structures of only about 15% of all the species in the family have been described. Some published descriptions included measurements of every structural parameter, while others only included partial information on melanosome modifications. For example, descriptions for only 61% of species included details about melanin layer thickness, while descriptions for almost all species had complete information on the presence/absence of melanosome hollowness and/or platelet shape (92%). Most species records (83%) described the type of structure (single-layered or photonic crystal). These data, though taxonomically biased, allowed us to describe the properties of the three melanosome modifications we defined (thin melanin layers, hollowness, and platelet shape). Using an avian phylogeny (*Jetz et al., 2012*), we mapped these modifications for all 280 species for which complete information on melanosome type was present in our database (*Figure 3A*). Although these species represent only a fraction of those with iridescent feathers, the major iridescent orders are represented. Our analysis thus provides a broad snapshot of iridescent feather structure diversity and evolution across birds. In the sections below, we use this data set to test functional hypotheses for each modification and to discuss evolutionary patterns in more detail.

## Thin melanin layers

We have suggested that all four melanosome types found in brilliant iridescent structures (*Figure 2B–E*) share a common trait: a reduction in melanin layer thickness. This is plausible based on the measurements and description of melanosome types given by *Durrer, 1977*, who proposed a division of solid rods into a thinner (diameter of ~100 nm) and thicker variety (diameter of ~200 nm), but has not been formally quantified. In the current literature, solid rods are often treated as a single melanosome type with a continuous size distribution (*Eliason et al., 2013*; *Maia et al., 2013b*; *Nordén et al., 2019*). Thus, to study the evolution of thin melanin layers, we first needed to explore the distribution of solid rod diameters using the feather iridescence database. Specifically, we used the feather iridescence database to show that: (1) solid rods can be divided into two distinct distributions (a thinner and thicker variety); and (2) hollow and/or platelet-shaped melanosomes have equally thin or thinner melanin layers than thin solid rods, demonstrating that they share this modification.

Analyzing the distribution of melanosome diameter in all solid rods, we found a significant bimodal distribution (*Figure 4*, unimodality rejected, $p<0.001$, bimodality not rejected, $p=0.888$). Based on the bimodal distribution of melanosome diameters in solid rods, we define 'thick solid rods' as those with a diameter ≥190 nm and 'thin solid rods' as those with a diameter <190 nm. It should be noted that this definition differs slightly from Durrer's categorization, which specifies a range of 70–140 nm for the thin solid rods he measured (*Durrer, 1977*). Thick solid rods are similar in size to melanosomes in black feathers (*Figure 4*; data from *Li et al., 2012*), supporting the hypothesis that thick solid rods represent minimally modified or unmodified melanosomes. Iridescent structures most likely evolved from black plumage (*Maia et al., 2012*; *Shawkey et al., 2006*); therefore, we can use the size of melanosomes in black feathers to represent an 'unmodified' melanosome. In contrast, the melanosomes in black feathers are considerably thicker than the thin solid rods in iridescent feathers (*Figure 4*), suggesting that thin solid rods are considerably modified from the ancestral state.

We can now define 'thin melanin layers' as any melanosome with melanin layers <190 nm. This value is just below the upper limit in our expected range for melanin layer thickness (206 nm), in line with our prediction (see *Identifying key melanosome modifications*). Using this new definition, we found that all hollow and/or platelet-shaped melanosomes can indeed be classified as having thin melanin layers (range 24–139 nm, *Figure 5*). Whether a single melanin wall in hollow melanosomes always represents one melanin layer is debatable: some photonic crystals with hollow melanosomes have little or no keratin interspersed between melanosome layers (e.g., *Figure 2C and E*). In these cases, it may be more appropriate to think of a single melanin layer as the sum of two melanin walls. However,

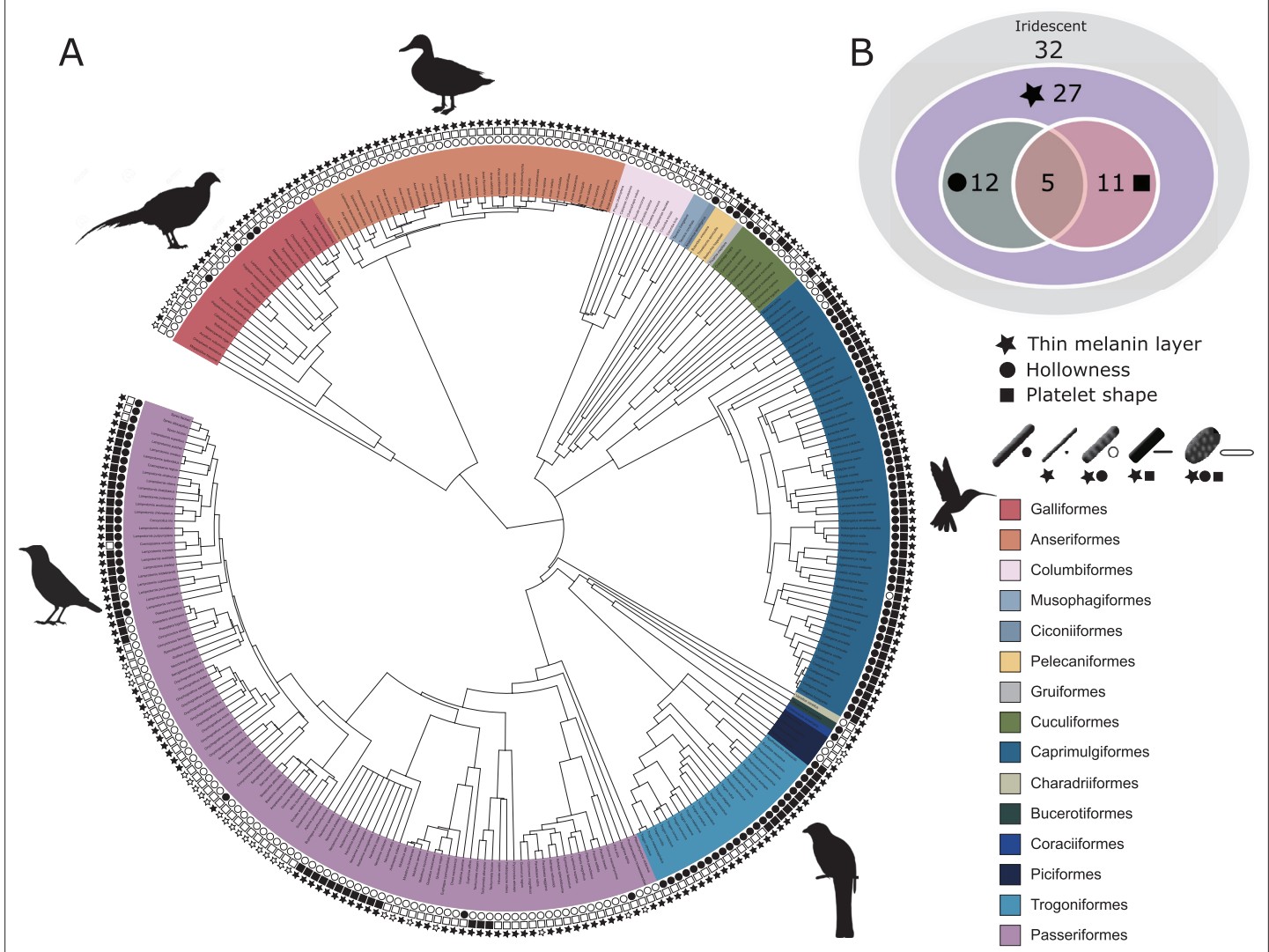

**Figure 3.** Evolutionary distribution of three key melanosome modifications in iridescent structures: thin melanin layers (star), hollowness (circle), and platelet shape (square). Schematics of melanosomes in the key show how combinations of modifications correspond to each melanosome type. (**A**) Melanosome modifications mapped onto a phylogeny including all species in the feather iridescence database (280 species, after excluding 26 species lacking data on melanosome type). Note that where data on melanin layer thickness was not available for a species with hollow and/or platelet-shaped melanosomes, they were assumed to have thin melanin layers, since all known hollow and platelet structures do. Silhouettes shown for the five families that are best represented in the feather iridescence database (>20 species represented in the database): Sturnidae, Phasianidae, Anatidae, Trogonidae, and Trochilidae. (**B**) Venn diagram showing the number of bird families in the feather iridescence database for which each modification was present. The majority of bird families with iridescent plumage studied have evolved thin melanin layers, and there are no hollow or platelet-shaped melanosomes that have not also evolved this modification. A similar number of families have hollow or platelet-shaped melanosomes, but only five families have evolved both modifications together. Note that this plot depicts the number of occurrences of each modification, not independent evolutionary origins. Silhouettes from Phylopic.org, licensed under a Public Domain License (full details in *Supplementary file 1o*).

all hollow forms in photonic crystals have a single melanin wall thickness of <95 nm (*Figure 5*), so they would still qualify as 'thin' even if this value were doubled. All four derived melanosomes with thin melanin layers have significantly thinner melanin layers than melanosomes in black feathers and thick solid rods (phylogenetic pairwise *t*-test, all *p*<0.01, see details in *Supplementary file 1a*).

Next, we tested our hypothesis that thin melanin layers evolved for a specific optical benefit—to allow photonic crystals to produce bright and saturated colors. We have already shown that the four derived melanosomes share the modification of thin melanin layers, but it is possible that this evolved for reasons unrelated to color production, such as to minimize the cost of melanin production. We predicted that if thin melanin layers did evolve for an optical benefit, they should have converged on

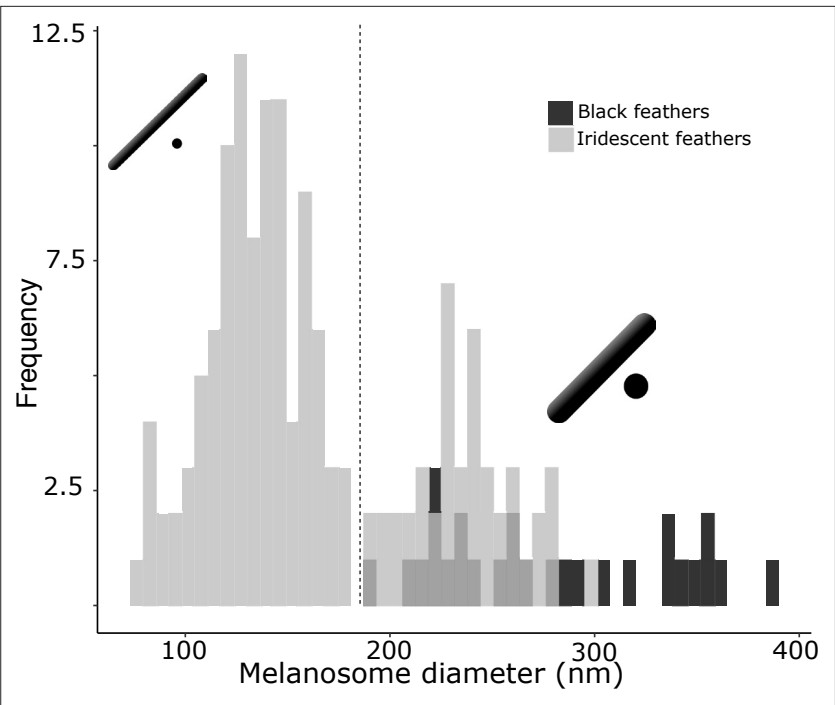

**Figure 4.** There are two distinct types of solid rods in iridescent structures: thick solid rods and thin solid rods. This is evident from the clear bimodal distribution shown by the histogram of melanosome diameters found among all solid rods in the feather iridescence database (gray). Based on this distribution, we define 'thin solid rods' as any solid rod with a diameter <190 nm (marked with dashed line). Plotted in black is the distribution of diameters from melanosomes in black feathers (data from **Li et al., 2012**), which overlaps with the distribution of thick solid rods in iridescent structures.

the expected range for producing bright interference peaks in the bird-visible spectrum (i.e., a layer thickness between 37.5–206 nm). In addition, we predicted that melanosomes with a thickness outside this favorable range should be rare or absent in photonic crystals. We found that all derived melanosomes indeed have converged on thicknesses well within this expected range (**Figure 5**). Moreover, all derived melanosome types achieve optical thicknesses of $\frac{\lambda}{4}$ (37.5–103 nm). Such structures could in theory produce ideal multilayers, which produce the greatest reflectance for a two-material reflector (**Land, 1972**). We also found that the vast majority of photonic crystals contain melanosomes with thin melanin layers (99% of all species with photonic crystals). Overall, these findings are compatible with the hypothesis that the primary benefit of thin melanin layers in photonic crystals is to produce bright and saturated colors. The importance of this key modification—thin melanin layers—for iridescent color production can also be inferred from its phylogenetic distribution. Over 80% of all families represented in the feather iridescence database have evolved thin melanin layers (27 out of 32 families, **Figure 3B**). The families that lack the thin modification also lack species with brilliant iridescent plumage (Numididae, Aegithinidae, Irenidae, Buphagidae, Megapodiidae, and Lybiidae).

However, it is also true that many single-layered structures are formed with melanosomes with thin melanin layers (present in 59% of all species with single-layered structures). If derived melanosome types evolved to provide brilliant iridescent color via thin melanin layers, why do they exist in single-layered structures, too? We propose two likely explanations. The first possibility is that thin melanin layers are advantageous also in single-layered structures, by forming a two-layered structure when the cortex is of a similar thickness to the melanin layer. This would increase the saturation and brightness of interference colors, though the effect would not be as strong as in a photonic crystal with many layers. The second possibility is that derived melanosomes evolve in single-layered structures due to another advantage, not because of their thin melanin layers. For example, a hollow interior or platelet may be beneficial to increase the brightness of the color. In either scenario, thin melanin layers are potentially co-opted in photonic crystals to produce brilliant iridescence, a possibility we explore in the Discussion.

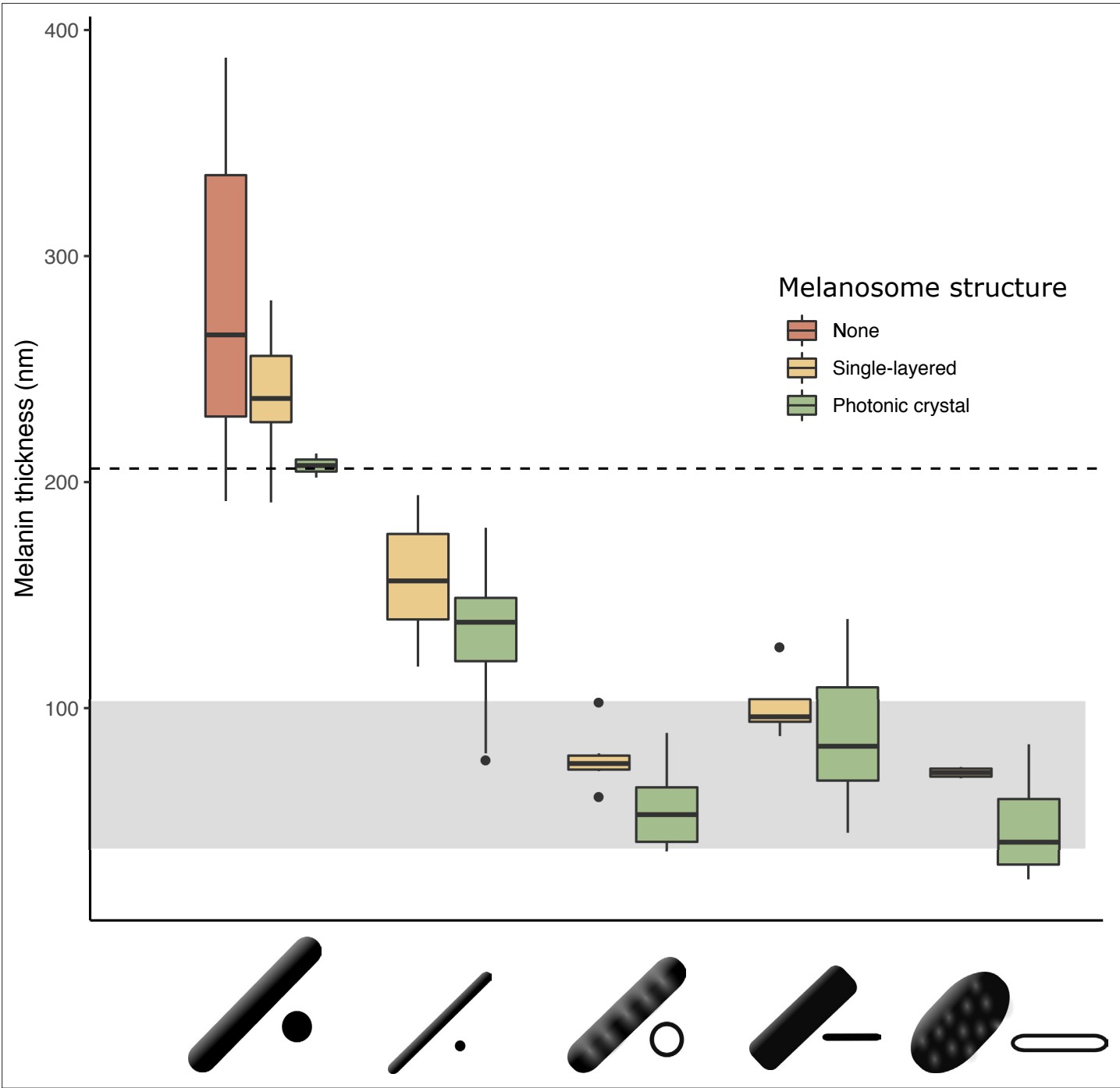

**Figure 5.** The thickness of melanin layers in derived melanosomes has converged toward the theoretical expected range, where optical thickness $<\frac{\lambda}{2}$ (below dashed line, for bird-visible spectrum). Boxplot shows the distribution of melanin layer thicknesses for each melanosome type in single-layered structures (yellow) and photonic crystals (green) in the feather iridescence database. 'None' corresponds to melanosomes in a black feather without organization (data from *Li et al., 2012*). All melanosome types except thick solid rods, which are predominantly found only in single-layered structures with weak iridescence, have converged toward an optical thickness of $<\frac{\lambda}{2}$. Hollow and platelet forms often reach thicknesses closer to $\frac{\lambda}{4}$, which can in theory form ideal multilayers (gray box, for bird-visible spectrum). Note that three species are recorded to have thick solid rods in a photonic crystal: *Paradisaea rubra* (Red bird-of-paradise), *Parotia lawesii* (Lawes' parotia), and *Eudynamys scolopaceus* (Asian koel). *Paradisaea rubra* and *Parotia lawesii* have melanosomes with porous interiors (see Discussion), so they are not 'true' thick solid rods and will have an optical thickness closer to that of a derived melanosome. The structure in *Eudynamys scolopaceus* consists of tightly, hexagonally packed rods and appears to produce relatively weak iridescent color. It may be an example of a structure evolving toward brilliant iridescence (Path 2, *Figure 7C*).

## Hollowness

Hollowness occurs in both rod-shaped and platelet-shaped melanosomes. However, whether the size of internal air pockets ($d_{air}$, Figure 8C) differs in hollow rods compared to hollow platelets has never been tested. If air pockets function primarily to produce strong interference colors in bird-visible wavelengths, we predict that there should be no difference between the air pocket diameters in hollow rods and hollow platelets and that diameters should be constrained between 75 and 350 nm (see **Identifying key melanosome modifications**). On the other hand, if hollowness evolved for different reasons in rods and platelets, and/or for non-optical functions, their air pocket diameters may differ. Air pocket diameter ranged from 50 to 251 nm and did not differ significantly between rods and platelets (phylogenetic ANOVA, $F$(1, 55)=16.80, $p$=0.176, $df$=1). This range does indeed include the thickness (75–350 nm) that would produce interference colors of the first order in the bird-visible range. Taken together with our results on melanin layer thickness (see previous section), both air pockets in hollow melanosomes and thin melanin layers appear to be tuned to produce bright and saturated colors.

Our phylogenetic analysis shows that a hollow interior has evolved in at least 12 bird families, or 34% of all families in the feather iridescence database (**Figure 3B**). Many families with brilliant iridescence are included, such as Phasianidae, Trochilidae, and Sturnidae. However, hollow melanosomes do not appear to be a requirement for brilliant iridescence. Unlike thin melanin layers, which are present in all families exhibiting brilliant iridescence, hollow melanosomes are absent in many families containing brilliant iridescent species, such as Nectariniidae, Paradisaeidae, and Columbidae. Still, the occurrence of a hollow modification is phylogenetically widespread. The 12 families with a hollow modification belong to 10 different orders (Galliformes, Coraciiformes, Passeriformes, Bucerotiformes, Trogoniformes, Cuculiformes, Pelecaniformes, Caprimulgiformes, Piciformes, and Ciconiiformes), which suggests that the genetic changes associated with producing a hollow melanosome are either likely to occur or are highly conserved in birds. A more comprehensive phylogenetic analysis will be required to determine how many times hollow melanosomes have evolved independently in birds, but our study indicates that this modification evolved many times independently.

## Platelet shape

We classified structures as 'platelet-shaped' if they diverged from a circular cross-section. The degree of divergence varies, resulting in platelets with a range of eccentricities. Unfortunately, with few exceptions, the studies surveyed did not include measurements of the width of platelets, preventing us from quantifying and exploring the eccentricity of platelets. We did not find support for the hypothesis that platelets allow birds to incorporate a greater number of layers in the iridescent structure. There was no significant difference between number of layers in structures with platelets compared to rods (phylogenetic ANOVA, $F$(1, 220)=21.88, $p$=0.321).

Platelets are present in 11 bird families, or 31% of all families represented in the feather iridescence database (**Figure 3B**). This is very similar to the frequency of the hollow modification (34% of families). In fact, many of the families that have evolved a hollow modification have also evolved platelets. In some cases, the modifications have evolved in combination, producing hollow platelets—but in other cases solid platelets and hollow rods have evolved separately within a family. Only Nectariniidae, Hirundinidae, Hemiprocnidae, Apodidae, and Psophiidae have evolved platelet shapes but never hollow forms (**Figure 3A**, with the caveat that this may change with increased sampling). As with hollowness, platelets are present in many but not all families with brilliant iridescence. For example, platelets are absent in Paradisaeidae, Phasianidae, and Columbidae. Nevertheless, platelets are widely distributed across birds; they are present in seven different orders (Passeriformes, Pelecaniformes, Caprimulgiformes, Trogoniformes, Gruiformes, Piciformes, and Cuculiformes).

## Evolution of multiple modifications

We hypothesized that hollow and platelet shape modifications are in fact different mechanisms for achieving thin melanin layers. This is supported by the fact that hollow and platelet-shaped melanosomes always have thin melanin layers—there are no platelets or hollow forms with melanin layers ≥190 nm. However, five bird families have evolved all three modifications: thin melanin layers, hollowness, and platelet shape (Trochilidae, Trogonidae, Sturnidae, Galbulidae, and Threskiornithidae, **Figure 3B**). If hollowness and platelet shape are alternative ways to achieve thin melanin layers, then

why have some birds evolved both? The repeated evolution of hollow platelets suggests that at least one modification carries some additional functional value. For example, hollowness may in itself also increase the brightness of colors. Though it is possible that both modifications evolved together due to a shared mechanistic path, rather than due to some adaptive benefit, this is unlikely because species in each order with hollow platelets have close relatives with solid platelets, solid rods and/or hollow rods (*Figure 3A*). Moreover, as noted above, some species within a family have evolved solid platelets while others have hollow rods. Thus, there does not appear to be a strong constraint on evolving these particular modifications together, since each modification exists in isolation.

## Optical consequences of modified melanosomes

To understand how each melanosome modification affects color production in brilliant iridescent structures, we simulated light reflection from different structures using optical modeling. We generated 4500 unique structures (900 for each of the five melanosome types) that varied systematically in structural parameters (including diameter of melanosomes, lattice spacing, hollowness, and platelet shape; see full model description in Materials and methods). All the structures were of photonic crystal type, since we were interested in the evolution of brilliant iridescence. The parameter ranges used to generate the structures were derived from the known ranges reported in the feather iridescence database for each melanosome type (*Table 1*). Thus, although the simulated structures are hypothetical, they represent a realistic approximation of the structural variation that could exist, while allowing us to standardize parameters that could bias comparisons in real structures. For example, we modeled all simulated structures with four layers (the median number of layers for photonic crystals in the feather iridescence database), while real structures have varying numbers of layers, which would affect the brightness and saturation of colors independent of melanosome type.

We modeled the simulated reflectance spectra in avian color space to estimate color saturation and diversity in a manner that is relevant to bird color perception. The avian tetrahedral color space represents all the colors a bird can theoretically perceive (*Endler and Mielke, 2005*; *Goldsmith, 1990*; *Stoddard and Prum, 2008*). Reflectance spectra can be represented in tetrahedral avian color space as a function of how they would stimulate a bird's four color cone types. Once reflectance spectra are mapped in avian color space, we can extract values of saturation (distance to the achromatic center of the tetrahedron) and color diversity (mean Euclidean distance between all points—color span, and number of voxels occupied, see Materials and methods for details). To quantify the brightness of a spectrum, we used two measures: (1) peak reflectance (% reflectance at the wavelength of maximum reflectance); and (2) estimated stimulation of the avian double cones, which may play a role in achromatic perception (*Hart, 2001*; *Jones and Osorio, 2004*). We refer to both metrics as 'brightness' for convenience; the term luminance is often used to describe the perception of signal intensity (here modeled using the avian double cones). Taken together, these metrics give a good representation of the saturation, color diversity, and brightness of simulated reflectance spectra, where saturation and brightness together describe the brilliance.

Optical modeling revealed that thick solid rods are severely constrained in color diversity (*Figure 6A*). The simulated spectra are clustered toward the center of the tetrahedron, which means that they are producing colors of low saturation. In known feather nanostructures, thick solid rods are almost exclusively found in single-layered structures, which produce colors of low saturation and brightness. In theory, low color saturation and brightness could be due to the single-layered structure, as opposed to the melanosome type. However, we modeled all structures with four layers, suggesting that it is the thick solid rods themselves—and not the number of layers—that limits color production. In other words, producing saturated colors is not possible with thick solid melanosomes, irrespective of whether the structure is single-layered or a photonic crystal.

In contrast, all four derived melanosome types with thin melanin layers are capable of producing a large range of saturated colors (*Figure 6B–E*). Color diversity (color span and voxel occupancy) is very similar for the four derived types, suggesting that melanin thinness—the only modification they all share—is the most important modification for achieving saturated and varied colors (*Figure 6K–L*). We note that the four derived melanosome types can also produce unsaturated colors, near the origin of the colorspace (*Figure 6B–E*). This is not surprising, since our simulated structures were generated across the full range of possible combinations of melanosome size and spacing. The reflectance from a photonic crystal will depend on both melanosome size and spacing. Thus, even if the melanin layer

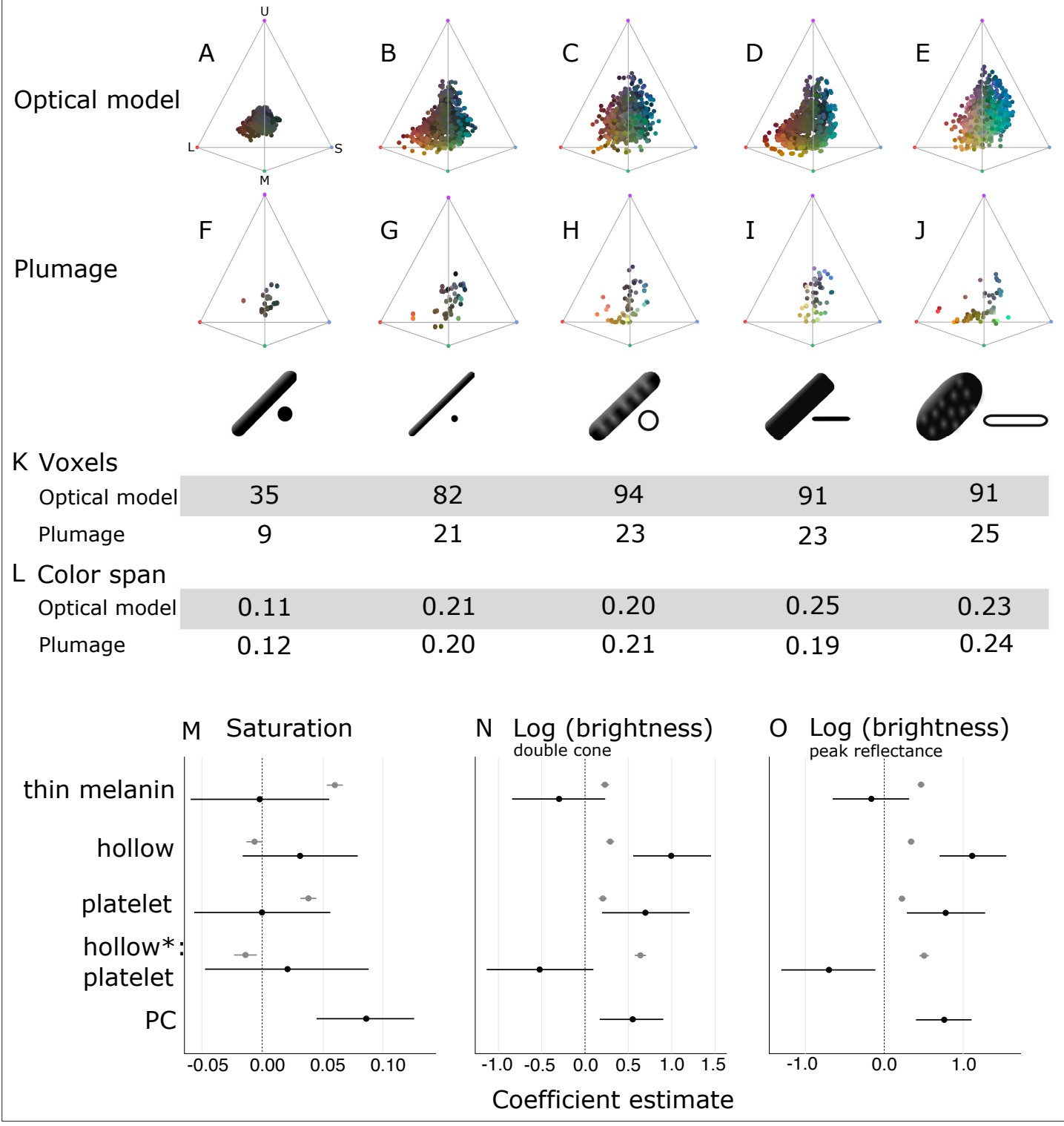

**Figure 6.** Optical effects of different melanosome modifications, as predicted by an optical model and found in empirical plumage analysis. (A–J) show the color diversity for structures with each type of melanosome represented in an avian tetrahedral color space (optical model (A–E); plumage data (F–J)). Statistics for color diversity are presented in terms of the number of occupied voxels (K) and mean color span (L) for both data sets. Thick solid rods produce colors of substantially lower diversity and saturation (A, F) than all melanosome types with thin melanin layers (B–E, G–J). In contrast, hollowness and platelet shape do not affect color diversity notably (C–E, H–J). (M–O) depict the estimates for the effects of each melanosome modification on saturation (M), log (brightness, double-cone) (N), and log (brightness, peak reflectance) (O), as predicted by linear models. The parameter PC describes variation explained by having a photonic crystal, which was used to control for variation in plumage data (see Results). Gray

*Figure 6 continued on next page*

*Figure 6 continued*

points show coefficient estimates for a model based on optical model simulations, and black dots show the posterior coefficient estimates for a model based on the plumage data. Horizontal lines show 95% confidence intervals for estimates.

The online version of this article includes the following figure supplement(s) for figure 6:

**Figure supplement 1.** Plumage color data (represented in avian color space) subdivided by type of structure; single-layered (**A–E**) and photonic crystal (**F–J**).

**Figure supplement 2.** Color saturation increases for structures with solid rods of diameter <190 nm.

thickness is within the expected range, it needs to be paired with the appropriate keratin thickness to produce bright interference colors (see ***Identifying key melanosome modifications***). Saturation will also vary for structures with thin melanin layers depending on how closely they approximate the ideal condition (where optical thickness of melanin layers = $\frac{\lambda}{4}$). This pattern can be seen by plotting mean saturation for simulated structures with solid rods of increasing diameters (***Figure 6—figure supplement 2***). There is no instantaneous leap to high saturation at 190 nm—rather, at 190 nm saturation starts to increase and reaches a peak at around 100 nm. This fits well with our theoretical expectation—saturation should start to increase below 206 nm (where optical melanin layer thickness $<\frac{\lambda}{2}$) and then peak around 103 nm (upper value where the condition of optical melanin layer thickness = $\frac{\lambda}{4}$ is satisfied).

To explore the effects of thin melanin layers, hollowness, and platelet shape on color properties in detail, we constructed linear models with melanosome modifications as binary predictors (present and absent) and saturation and brightness (described by two measures: double cone stimulation and peak reflectance) as responses. This allowed us to separate the effects of the different modifications, which are combined in many melanosome types (e.g., hollow rods are both hollow and have thin melanin layers). In agreement with the results for color space occupancy (***Figure 6A–E***), melanin layer thickness explained the greatest amount of variation in saturation in our linear model (***Figure 6M***; gray points). A positive effect was also seen for a platelet shape, which suggests that of the four derived melanosome types, solid platelets produce colors of the highest saturation. Small losses in saturation are incurred from incorporating hollowness, as can be seen from the negative coefficients of the variable hollowness and the interaction term hollowness×platelet shape (describing hollow platelets).

The linear model yielded similar results for both of our brightness measures (***Figure 6N–O***; gray points). All modifications increase brightness, but this effect is strongest for the interaction of hollowness and platelet shape (***Figure 6N–O***; gray points). Thus, the optical model predicts that hollow platelets produce the brightest colors. This effect likely arises from a lowered overall refractive index of melanosome layers with hollow platelets, which have a lower melanin-to-air ratio than layers built with hollow rods. However, this effect may be considerably weaker in real structures, where hollow platelets often have an internal honeycomb-like structure of melanin (***Figure 2E***), which would make the effective refractive index closer to that of hollow rods. Thick solid rods, hollow rods, and solid platelets produce colors that are less bright than those of hollow platelets but similarly bright to one another (***Figure 6N***; gray points).

Taken together, these results indicate that evolving thin melanin layers is the single most important factor for dramatically increasing color diversity and saturation (***Figure 6A–E and M***; gray points) while simultaneously increasing brightness (***Figure 6N–O***; gray points). When the effect of thin melanin layers is accounted for, a platelet shape has a similar but weaker effect on saturation and brightness (***Figure 6M–O***; gray points). Hollowness only increases brightness further (***Figure 6M–O***; gray points).

## Testing predictions with plumage data

Next, we investigated whether we could recover the same patterns in the iridescent plumage of birds with different nanostructures. We collected spectral data from 111 patches on 80 species that were represented in the feather iridescence database and possessed known melanosome types. Plumage patches included weak and brilliant iridescent colors, with melanosomes arranged in single layers and photonic crystals, respectively. We included single-layered structures in our plumage data set since thick solid rods do not occur in photonic crystals. Including single-layered structures allowed us to compare all five melanosome types.

In agreement with the optical model results, the color diversity of structures with thick solid rods is low, almost half of that found in structures with thin melanin layers (*Figure 6F*). Moreover—mirroring the results in our optical model simulations—thin solid rods, hollow rods, solid platelets, and hollow platelets are all nearly equal in color diversity (*Figure 6K–L*). While the four derived melanosome types also produce unsaturated colors, these colors are mainly produced by single-layered structures (*Figure 6—figure supplement 1*). However, some differences between the optical model simulations and plumage data are noteworthy. In contrast to other melanosome types, solid platelets do not produce any saturated red colors. This is unlikely to be due to any inherent developmental or physical constraint, since our optical model simulations—based on realistic melanosome properties, including size—indicate that solid platelets can clearly produce colors in this area of color space (*Figure 6D*). Rather, this effect may be a consequence of phylogenetic bias, as the majority of species with solid platelets in our data set are sunbirds (Family Nectariniidae), a group that uses carotenoid pigments—rather than structural colors—for red plumage coloration.

To explore further how thin melanin layers, hollowness, and a platelet shape affect saturation and brightness, we fitted generalized linear mixed models using Bayesian methods that allowed us to account for multiple measurements within a species (i.e., we obtained two reflectance measurements per plumage patch per species). In contrast to our optical model simulations, melanosomes in the real plumage patches we measured were arrayed in a variable number of layers. Since having many layers is known to increase the brightness and saturation of colors, we added a parameter to control for this effect. The binary parameter 'PC' (photonic crystal) described whether a structure contained a single layer of melanosomes (not a photonic crystal), or several repeating layers of melanosomes (photonic crystal). In our data set, 38 species had a single layer and 42 species had a photonic crystal structure.

In this linear model, there were no significant effects of either platelet shape or hollowness on saturation (*Figure 6M*; black points). Thus, we did not find support for the optical model prediction that solid platelets produce more saturated colors. We also did not find a significant positive effect of thin melanin layers (*Figure 6M*; black points), in contrast to our findings with the optical model. However, since our plumage data did not include any photonic crystals with thick melanin layers, the effect of thick versus thin melanin layers could only be compared for single-layered structures. Thus, our model suggests that thin melanin layers do not increase saturation for *single-layered* structures. This is supported by the low and similar color diversity seen across single-layered structures, irrespective of melanosome type (*Figure 6—figure supplement 1*). The model also confirms that photonic crystals produce colors of significantly higher saturation than single-layered structures (*Figure 6M*; black points). While the plumage data cannot directly tell us the effect of thin melanin layers in a photonic crystal, the simulations from our optical model show that thick solid rods in a photonic crystal would not produce more saturated colors than a typical single-layered structure (*Figure 6A*). Thus, the plumage data and optical model together suggest that *both* a photonic crystal and thin melanin layers are required to produce saturated and diverse colors.

In terms of brightness, the plumage data compare to the optical model simulations in interesting ways. In agreement with the optical model, the linear model revealed a significant positive effect of hollowness and platelet shape on the brightness of colors (*Figure 6N–O*; black points). However, we did not see a large positive effect of hollow platelets in the empirical data. In fact, this parameter has a negative effect, which is significant for peak reflectance (*Figure 6O*; black points). This discrepancy may be due to the fact that—in the real plumage structures measured—hollow platelets tended to be arranged in relatively few layers. Our sample of structures with solid platelets consisted almost entirely of different species of sunbirds (Family Nectariniidae), which exhibit 5–8 layers (*Durrer, 1962*), while the sample for hollow platelets contained several groups with fewer layers (e.g., *Durrer, 1977*). We could not control for this because the number of layers is not known in many of the structures we sampled; instead, we only included a parameter to indicate if a structure was a photonic crystal or not. We can, however, compare the brightness of *single-layered* structures with hollow platelets versus solid platelets. This comparison shows that the hollow platelets produce brighter colors (phylogenetic ANOVA, $F(1, 34)=12.10$, $p=0.034$, $df=1$). Thus, the general conclusion that hollowness increases brightness is well supported, although this advantage is likely to diminish with increasing number of layers in the structure. Reflection from a multilayer with melanin and keratin becomes saturated at >9 layers (*Land, 1972*), so it is likely that the greatest advantage of hollowness is gained for structures with ≤9 layers.

When interpreted alongside the optical modeling, the plumage data support the general conclusions that thin melanin layers, in combination with a photonic crystal structure, are critically important for producing diverse and brilliant colors, while hollowness and platelet shape are less crucial. We observe a near doubling of color diversity for real plumage structures with thin melanin layers compared to structures with thick solid rods, consistent with the results of the optical model. While the plumage data alone cannot prove that this difference is driven by the combination of thin melanin layers and photonic crystal structure, as opposed to the PC parameter by itself, our optical models exclude this possibility (see *Figure 6A* for a simulation of photonic crystals with thick solid rods). Hollowness and a platelet shape increase the brightness of colors further, in agreement with the optical model.

## Discussion

Brilliant iridescence has been linked to the evolution of different melanosome modifications, most notably hollowness and a platelet shape (*Eliason et al., 2013*; *Maia et al., 2013b*), but how these modifications affect color production has not been evaluated in a unified framework. Here, we have taken a broad approach, comparing all five melanosome types found in iridescent feathers, to uncover general design principles governing the production of brilliant iridescence. We find that the most important modification for increasing brilliance is not hollowness or a platelet shape *per se*, but rather a third modification that unites all melanosomes found in brilliant iridescent structures: thin melanin layers. Specifically, we show that melanosomes in brilliant structures have converged on a melanin layer thickness of approximately 40–200 nm (*Figure 5*), which is the theoretical expected thickness to produce first-order interference peaks in the bird-visible spectrum. Our optical simulations and empirical data demonstrate that this modification alone nearly doubles color diversity (*Figure 6A–L*) and simultaneously increases saturation and brightness (*Figure 6M–O*). In contrast, hollowness and platelet shape on their own only contribute to increased brightness, in line with earlier work on hollow rods (*Eliason et al., 2013*).

Our results have interesting implications for the evolution of brilliant iridescent structures in birds. For the production of weakly iridescent colors, it is sufficient to organize a single layer of melanosomes of any size, since it is typically the thickness of the overlying keratin cortex that controls the interference color (*Doucet et al., 2006*; *Maia et al., 2009*). In contrast, to produce brilliant iridescence, we show that two key optical innovations are required: a photonic crystal (multiple periodic layers of melanosomes) and melanin layers with an optical thickness $<\frac{\lambda}{2}$. Indeed, *Durrer, 1977* observed that these two features were common to the brilliant structures he studied and here we validate the importance of his observation with optical modeling and plumage color measurements. Specifically, we find that saturation increases for structures with melanin layer thickness <190 nm (*Figure 6A–E*, *Figure 6—figure supplement 2*), which we define as 'thin melanin layers.' Above this value, iridescent structures produce colors that have low saturation and brightness, irrespective of the number of melanosome layers (*Figure 6A*). This insight could be used to place a lower bound on when brilliant iridescence first evolved in feathers, using the fossil record. For example, the preserved melanosomes from the plumage of *Microraptor*, a feathered theropod that is predicted to have exhibited iridescent plumage, have an average diameter of 196 nm (*Li et al., 2012*). This suggests that *Microraptor* exhibited weak iridescence, as opposed to brilliant iridescence. However, we caution that preserved melanosomes with melanin layers <190 nm do not necessarily prove that the feathers originally produced brilliant iridescence. Both a photonic crystal and thin melanin layers are required to produce brilliant iridescence—and the three-dimensional structure is usually lost in fossil feathers.

Our results show that photonic crystals with all four melanosome types found in brilliant iridescent structures have similar optical qualities. This suggests that variability in melanosome type may be strongly influenced by historical factors, as opposed to particular types being associated with specific optical functions. In other words, birds have a seemingly flexible 'nanostructure toolkit' with which to produce diverse and brilliant iridescent colors. Thus, the reason that sunbirds (Nectariniidae) produce brilliant iridescence with solid platelets while hummingbirds (Trochilidae) mainly use hollow platelets (*Figure 3A*) is likely related to variation in evolutionary history rather than to variation in selection for different optical properties. Supporting this interpretation is the fact that diverse photonic crystals in birds often have independent evolutionary origins. In Galliformes, some families have photonic crystals with thin solid rods and others have photonic crystals with hollow rods (*Figure 3A*), but these different structures have almost certainly evolved from an ancestor with a non-iridescent or single-layered

structure rather than a photonic crystal (*Gammie, 2013*). Similarly, in Sturnidae, photonic crystals with hollow rods in the genus *Cinnyricinclus* and photonic crystals with hollow platelets in the genus *Lamprotornis* likely evolved independently from non-iridescent structures (*Figure 3A*, *Durrer and Villiger, 1970*; *Maia et al., 2013b*).

Yet in some groups, melanosome type is highly variable within the same genus, or even within the same species (interpatch variability). In the birds-of-paradise (Paradisaeidae), which typically display photonic crystals with thin solid rods, two species (*Paradisaea rubra* and *Parotia lawesii*) are known to have evolved large rods with a porous interior (*Figure 5*, *Durrer, 1977*; *Stavenga et al., 2015*). In Lawes' parotia (*Parotia lawesii*), other iridescent patches contain structures with thin solid rods, proving interpatch variability in melanosome type. Hummingbirds, whose iridescent structures are typically built with hollow platelets, can also exhibit interpatch variability in melanosome type. Some patches may contain a structure with solid platelets, or even mixed structures with both hollow and solid platelets (*Gruson et al., 2019*). It is notable that the only known examples of interpatch variability in melanosome type come from the birds-of-paradise and hummingbirds—groups that are known to have exceptionally high rates of color evolution (*Beltrán et al., 2021*; *Eliason et al., 2020*; *Ligon et al., 2018*; *Parra, 2010*). One hypothesis to explain this variation could be that modifications in hollowness/platelet shape tune the brightness of some patches (*Figure 6N–O*). However, this seems unlikely. Both birds-of-paradise and hummingbirds typically have >9 melanosome layers in their iridescent structures, which already achieves nearly 100% reflectance irrespective of melanosome type. Moreover, our results suggest that there would be little or no difference in brightness between structures with solid platelets and hollow platelets (*Figure 6N–O*)—only between thick solid rods and hollow and/or platelet-shaped melanosomes. Indeed, *Gruson et al., 2019* found color production to be similar among patches with different melanosome types in hummingbirds. We speculate that high interpatch variability in melanosome type in hummingbirds and birds-of-paradise is not related to general optical benefits of specific melanosome types or modifications, but rather to general high rates of color change in these groups (*Eliason et al., 2020*; *Parra, 2010*). Our optical modeling results (*Figure 6B–E*) show that there are multiple ways to reach the same areas of color space—using different melanosome types. It is possible that a change in melanosome type may be the fastest route to a new area of color space, even though the same color shift could in theory be produced by adjusting the size of the original melanosome type. This idea is hard to test with our current very limited understanding of the genetics of iridescent structures, but we predict that groups with high variation in melanosome type will have a greater standing variation in genetic traits associated with different melanosome types. We also predict that plumage patches with higher rates of color evolution will have greater variability in melanosome type.

We have proposed that evolutionary history can explain the diversity of derived melanosome types in iridescent feather nanostructures, as opposed to particular types being associated with specific optical functions. However, we cannot fully exclude hypotheses based on general adaptive explanations tied to melanosome type. Our plumage color data set, though phylogenetically broad, is relatively small, and it is possible that increased sampling could reveal some differences in color production between derived melanosome types. It would be important to pair a larger data set with detailed information on the number of layers in the iridescent structures, since this is a parameter for which we could not fully control (number of layers is rarely reported in the literature). Future studies should also investigate potential interactions with the feather micro- or macrostructure. Iridescent feathers are known to have highly modified barbules (*Durrer, 1977*), which has been shown to affect coloration in Lawes' parotia (*Stavenga et al., 2010*) and the African emerald cuckoo (*Chrysococcyx cupreus*, *Harvey et al., 2013*). The interaction of feather microstructure and coloration is an active field of study (*McCoy et al., 2021*), but how nanostructures and microstructures may interact is a largely unexplored topic. Such an investigation would likely explain some of the discrepancies between the plumage data (which is measured from many feathers with micro- and macro-shape) and the optical model simulations (which consider only the nanostructure). In addition, it is important to stress that we still lack a full understanding of how modified melanosomes function in single-layered structures. We found that hollow melanosomes and platelets only increase brightness, while saturation remained low irrespective of melanosome modifications in single-layered structures. However, our analysis did not investigate potential interactions between cortex thickness and melanosome type. Such interactions are of less importance in photonic crystals but could be significant for single-layered structures.

Cortex thickness and melanosomes could be tuned together to produce in effect a multilayered structure, which would result in a stronger interference peak. Such 'cortex tuning' may explain why some African starlings with a single layer of hollow platelets produce unusually bright and saturated colors (*Figure 6—figure supplement 1*). A more detailed investigation is key to understanding why the derived melanosome types are found not only in photonic crystals—where they produce brilliant iridescence—but also in weakly iridescent single-layered structures.

Another open question involves the potential non-signaling functions of different melanosome types. Melanin has been shown to influence a feather's mechanical properties (*Burtt, 1979*) and ability to resist bacterial degradation (*Goldstein et al., 2004*). An interesting question is therefore whether different types of iridescent structures contain different amounts of melanin. Our results show that structures with derived melanosomes have converged on a shared range of melanin layer thicknesses (*Figure 5*), which suggests that differences in melanin content may not be large. However, melanin content has never been compared across different iridescent structures. This would be an exciting avenue for future research, especially since differences in melanin production may induce pleiotropic effects on other traits, such as immune function or behavior (*Ducrest et al., 2008*).

Beyond elucidating the many functions of iridescent colors in birds, we need to understand how brilliant structures evolve to resolve fully the mystery of their structural diversity. To our knowledge, no general models have been proposed to explain how photonic crystals with modified melanosomes evolve from more simple, single-layered structures (but see discussion by *Durrer, 1977*; *Durrer and Villiger, 1970*). We can use the insights derived from our study to propose two hypothetical routes to brilliant iridescence.

Brilliant iridescent structures likely originated from single-layered structures with thick solid rods (*Maia et al., 2012*; *Shawkey et al., 2006*). To achieve brilliant iridescent colors, such a structure must evolve to incorporate a photonic crystal-like organization of melanosomes—and the melanosomes must have thin melanin layers. However, our results showed that either of these changes on their own does not increase color saturation or brightness. This leads to an interesting problem, where only the two adaptations *together* produce a great advantage in brilliance. How could such a structure evolve? We propose two evolutionary paths through which this may have occurred—either via elaboration of melanosome shape *first* and photonic crystals *second* (Path 1) or via elaboration of photonic crystals *first* and melanosome shape *second* (Path 2). Both paths lead to feather structures with thin melanin layers, fully capable of making a broad range of brilliant iridescent colors (*Figure 7*).

In the first route, modified melanosomes with thin melanin layers evolve for reasons unrelated to color saturation (*Figure 7B*), perhaps to enhance brightness. Hollow and platelet-shaped modifications may evolve initially to produce brighter colors, while thin solid rods have been hypothesized to facilitate the formation of thin film structures through their elongate shape (*Maia et al., 2012*). Once evolved, melanosomes with thin melanin layers allow for the evolution of photonic crystals, since such structures would produce brighter and more saturated colors. The second route to brilliant iridescence involves the spontaneous formation of a photonic crystal from a single-layered structure, which then selects for modified melanosomes with thin melanin layers (*Figure 7C*). In many single-layered structures, a discontinuous second layer can be seen beneath the top layer, where melanosomes are packed hexagonally (e.g., *Figure 2A*). This likely provides a more mechanically stable configuration during barbule development (as suggested by *Eliason et al., 2013*). It is easy to see how the evolution of hollowness in such a structure would lead to the production of brilliant iridescence.

The feather iridescence database gives some support to both of these hypothetical paths. Single-layered structures with modified melanosomes are relatively common (*Figure 5*), suggesting that Path 1—where melanosome shape diversifies first—may be a common route to more complex structures. In support of Path 2, hexagonally arranged photonic crystals with hollow rods are common in many groups (Galliformes and Trogoniformes) that also contain taxa with single-layered structures of thick solid rods. However, very few clades are sampled in sufficient detail to draw inferences about the transitions between different structures. To test our hypotheses, careful characterization of nanostructures in a group with repeated transitions to brilliant iridescence is needed. Such a study could also lay the groundwork for exploring the genetic regulation of iridescent structures, an area of research in its infancy (*Saranathan and Finet, 2021*; *Rubenstein et al., 2021*; *Price-Waldman and Stoddard, 2021*).

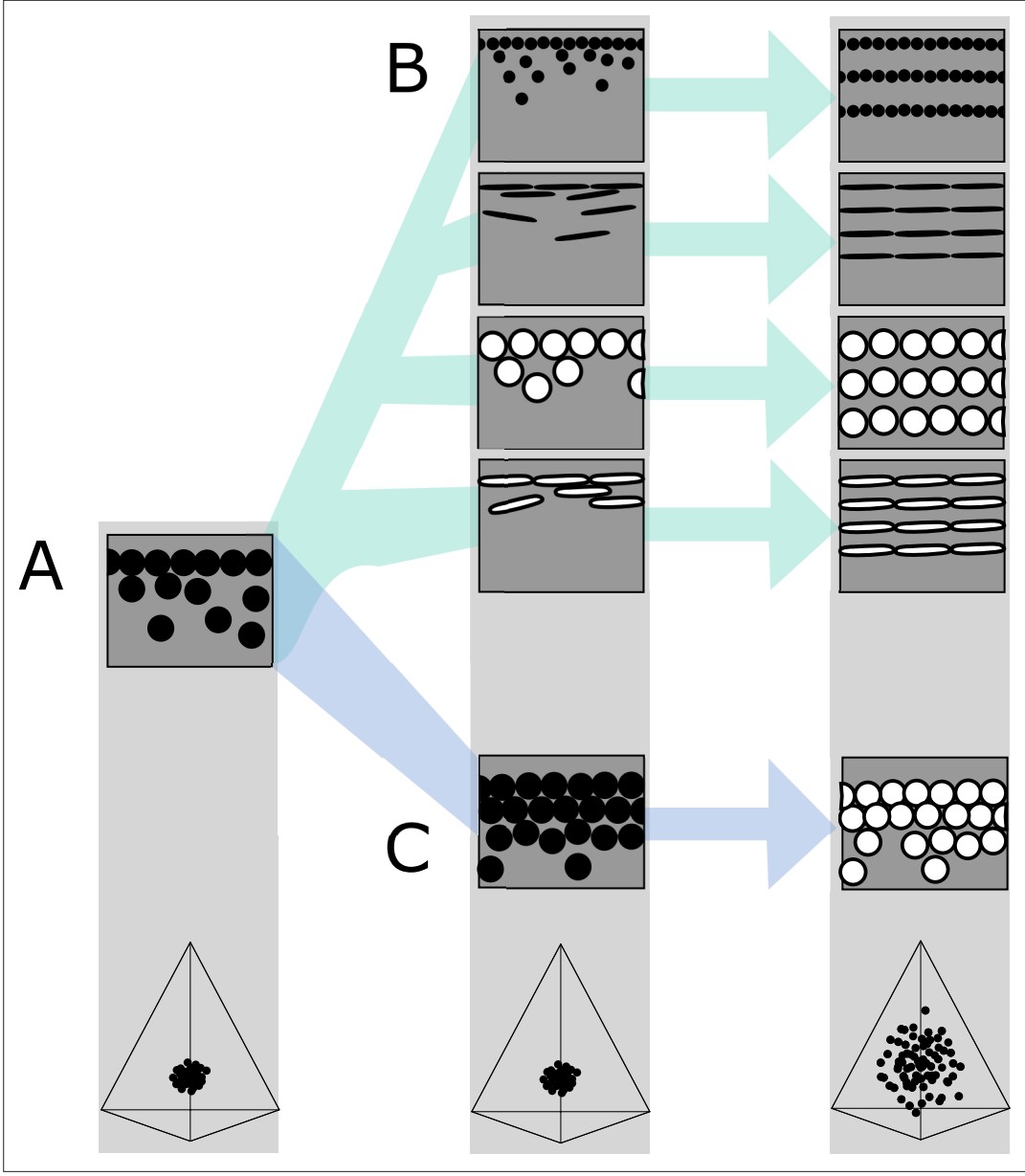

**Figure 7.** Hypothetical evolutionary paths to brilliant iridescence. Gray squares depict schematics of barbule cross-sections, showing the iridescent nanostructures within, while the tetrahedra below show hypothetical color diversity for each evolutionary 'step,' represented in avian color space. (**A**) Assumed ancestral state for iridescent structures—a single-layered structure with thick solid rods. Note that a layer refers to a *continuous* layer of melanosomes; scattered or disorganized melanosomes often seen below a continuous single layer do not constitute additional layers. From this state, structures may either first evolve modified melanosomes in a single-layered structure (**B**, Path 1) or first evolve multilayered, hexagonal structuring of thick solid rods (**C**, Path 2). Both of these states are expected to give a negligible advantage in terms of color saturation and diversity, as seen in the hypothetical color spaces corresponding to each stage (bottom). We argue that Path 1 might initially be driven by selection for brighter colors, while Path 2 could form spontaneously from higher concentrations of melanosomes in the barbule. Both paths can then evolve toward more brilliant forms (multilayers in (**B**), modified melanosomes with thin melanin layers in (**C**)), which will drastically expand the possible color diversity.

By investigating the evolution and optical properties of brilliant iridescent feather nanostructures spanning 15 avian orders, we have identified some features common to iridescent nanostructure design and some features that are likely to result from differences in evolutionary history. The key feature uniting melanosomes in brilliant iridescent structures is the presence of thin (<190 nm) melanin

layers, which tunes a photonic crystal to produce bright and saturated colors in the bird-visible spectrum. We suggest that much of the diversity in melanosome type in brilliant iridescent structures—such as the prevalence of solid platelets in sunbirds but hollow platelets in hummingbirds—could be explained by differences in evolutionary history, since different melanosome types offer alternative routes to producing thin melanin layers. We propose two likely evolutionary routes, which could be explored further in a careful study of a clade with repeated transitions to brilliant iridescence. This would clarify the steps associated with the evolution of brilliant iridescence—and potentially link these steps to genetic changes.

The large-scale patterns uncovered in this study are only a first step toward gaining a deeper understanding of how brilliant iridescence has evolved in birds. By focusing on large-scale patterns and general themes, our study may obscure or overlook some unique or unusual nanostructural strategies evolved by particular species or genera. However, our broad study should provide a powerful springboard for more focused studies.

## Materials and methods
### Building the feather iridescence database

We surveyed the literature for microscopy studies of iridescent feathers using two complementary approaches. For studies published earlier than 2006, we used the references in *Prum, 2006* and *Durrer, 1977* as a starting point. For later publications, we used Google Scholar to search for articles containing the terms 'iridescence' and 'feather.' We then extracted the following information from each study (where available, or possible to infer from redundant measurements): melanosome arrangement (single-layered, photonic crystal), melanosome type (i.e., solid rod, hollow rod, solid platelet, or hollow platelet), melanosome diameter ($d_{melsom}$), lattice spacing (a), the number of melanosome layers (n), diameter of hollow interior (if present, $d_{air}$), thickness of keratin layers (ks), thickness of melanin layers (mt; for solid forms $mt=d_{melsom}$, for hollow forms $mt=(d_{melsom}-d_{air})/2$), cortex thickness (c), the patch from which the studied feather originated, and the color of the feather. A schematic of all measurements is shown in *Figure 8*. With few exceptions, most studies sampled only a single iridescent patch from each species. This is based on the assumption that iridescent nanostructures are similar in all iridescent patches in a species, which seems to be true in most species but not all; hummingbirds and birds-of-paradise are the only known exceptions (*Durrer, 1977*; *Gruson et al., 2019*).

For a small number of records (n=17), we produced new measurements of iridescent structures using transmission electron microscope images previously collected by *Nordén et al., 2019*. Images were measured using the program ImageJ (*Abràmoff et al., 2004*). All images used for new measurements are available to download from the Dryad Digital Repository (doi.10.5061/dryad.4j0zpc8bq).

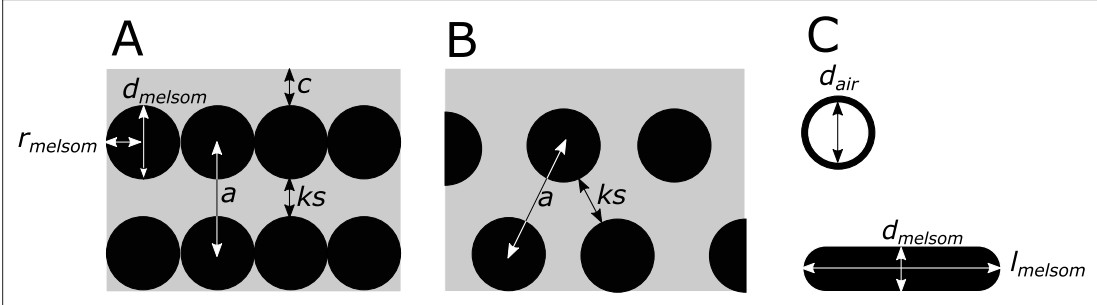

**Figure 8.** Definitions of parameters used in the study, shown in schematics of cross-sections of iridescent structures. (**A**) Laminar photonic crystal (multilayer); (**B**) hexagonal photonic crystal; and (**C**) isolated hollow and flat melanosomes. In (**A**), $d_{melsom}$: diameter of melanosome (shortest axis in flat melanosomes), $r_{melsom}$: radius of melanosome ($d_{melsom}$ /2), c: thickness of keratin cortex, a: lattice spacing (center-center distance between melanosomes), ks: keratin spacing (thickness of keratin layer between melanosomes at the thinnest point). In (**B**), keratin spacing (ks) and lattice spacing (a) are shown for a hexagonal photonic crystal. In (**C**), $d_{air}$: diameter of internal air pockets (shortest axis of air pockets in hollow platelets), $l_{melsom}$: width of platelets. Melanin layer thickness is defined as ($d_{melsom}$ for solid forms, and ($d_{melsom}-d_{air}$)/2 for hollow forms).

In total, our database covers 47 studies from 1952 to 2020 and 306 unique species, across 35 families and 15 orders (37% of total orders and 14% of total families in Aves, after taxonomy in *Billerman et al., 2020*. Out of these, 280 species had enough data to be included in our phylogenetic analyses.

## Phylogeny

We used the phylogenies of *Jetz et al., 2012*, which are based on a *Hackett et al., 2008* backbone, to construct a tree including all the species in the feather iridescence database and the species from the *Li et al., 2012* data set. The data set from *Li et al., 2012* of melanosome diameters in black feathers was included to compare with the data on melanosomes in iridescent structures. We sampled 1000 pruned trees from the tree distribution available at birdtree.org and then constructed a 50% consensus tree from this distribution. Branch lengths were calculated using the 'consensus.edge' function in the R package *phytools* (*Revell, 2012*). This tree was then pruned as necessary for different analyses.

## Optical modeling

We modeled the reflectance from iridescent feather structures using the software package MIT Electromagnetic Equation Propagation (MEEP) (*Oskooi et al., 2010*). Simulations were performed in one unit cell, with an absorbing perfectly matched layer in the $x$-direction, and periodic boundaries in the $y$-direction. Resolution was set to 80 pixels/μm, which gives 12 sampling points for one 300 nm wave in the material with the highest refraction index (melanin). We set the extinction coefficient ($k$) of melanin to 0.1, the refractive index ($n$) of keratin to 1.56, and the refractive index of melanin to 2. These values are an approximation based on published values (*Brink and Berg, 2004*; *Stavenga et al., 2015*). In reality, these values vary over the light spectrum for most materials. Both $n$ and $k$ decrease from short wavelengths to longer wavelengths for melanin. A higher refractive index is expected to broaden and increase reflection peaks, while a high extinction coefficient will tend to decrease the amplitude of the reflectance peak. Thus, peaks in the short wavelengths will tend to be slightly broader (but not taller), resulting in a brighter but less saturated color, compared to long-wavelength colors. However, we did not observe any large differences between modeled and plumage data in this direction, and thus we expect the effects of a varying refractive index to be insignificant for the larger patterns we describe. The extinction coefficient for keratin is likely to be low ($k$=0.03, *Brink and Berg, 2004*) and was omitted (set to 0).

The structural parameters varied in the model were melanosome diameter ($d_{melsom}$), relative hollowness ($d_{air}/d_{melsom}$), flatness ($l_{melsom}/d_{melsom}$), relative lattice spacing ($r_{melsom}/a$), and cortex thickness ($c$, *Figure 8*). We set the ranges for parameters related to melanosome shape to match the known ranges for each melanosome type, extracted from the feather iridescence database. For lattice spacing and cortex thickness, we modeled values over the total range reported in the feather iridescence database, and number of layers was fixed to 4 (the median in the feather iridescence database). For structures with rods, we modeled structures with a hexagonal packing in addition to the standard laminar configuration (*Figure 8B and A*, respectively) to represent the diversity present in real structures. Although a square configuration also exists, we did not model this since it has only been recorded in a single genus, the peafowls (*Pavo*). *Table 1* gives a detailed overview of the model settings for each melanosome type. Notice that the melanosome diameter of solid forms is varied in 30 steps, while

**Table 1.** Model parameter ranges for each melanosome type.

The values reported in parentheses are the number of evenly spaced steps with which the parameter was varied. For each melanosome type, we simulated 900 unique structural configurations.

| Melanosome type | Melanosome diameter (nm) | Hollowness ($d_{air}/d_{melsom}$) | Flatness ($l_{melsom}/d_{melsom}$) | Relative lattice spacing ($r_{melsom}/a$) | Cortex (nm) | Hexagonal packing |
|---|---|---|---|---|---|---|
| Thick solid rods | 190–300 (30) | 0 | 1 | 0.15–0.5 (5) | 5–1000 (3) | Yes |
| Thin solid rods | 65–180 (30) | 0 | 1 | 0.15–0.5 (5) | 5–1000 (3) | Yes |
| Hollow rods | 135–440 (10) | 0.26–0.69 (3) | 1 | 0.15–0.5 (5) | 5–1000 (3) | Yes |
| Solid platelets | 45–140 (30) | 0 | 2.4 | 0.15–0.5 (5) | 5–1000 (3) | No |
| Hollow platelets | 135–280 (10) | 0.26–0.69 (3) | 2.4 | 0.15–0.5 (5) | 5–1000 (3) | No |

the diameter of hollow forms is only varied in 10 steps. The thickness of melanin layers is important for determining hue, and hollow forms have two parameters that adjust this value (diameter and hollowness), while solid forms have only one (diameter). To avoid a bias toward greater hue variability in hollow forms due to this effect, we allowed the diameter of solid forms to vary in an equal number of steps as the combined effect of diameter and hollowness in hollow forms (10×3=30).

In total, we ran 4500 simulations, with 900 simulations for each melanosome type.

## Plumage measurements and spectral analysis

We collected spectral measurements of 80 bird species (across 13 orders) for which nanostructures were already known (see references in the feather iridescence database), housed in the American Natural History Museum, New York. Two individuals were used for each species, and all iridescent patches with different color (as perceived by human vision) were measured. In total, 111 unique patches were measured. Spectral measurements were taken directly on the specimen following standard procedures (*Andersson and Prager, 2006*). Briefly, we used a USB4000 spectrophotometer and a PX-2 xenon light source (Ocean Optics, Dunedin, FL). We measured color over a range of angles (15–135°) using a goniometer, keeping the light source fixed at 75°.

Spectra were analyzed in R v.3.6.1 (*R Development Core Team, 2019*) using the package *pavo* (*Maia et al., 2013a*). All spectral data were first smoothed to remove noise, using locally weighted smoothing (LOESS) and a smoothing parameter of 0.2. We removed negative values by adding the minimal reflectance to the spectrum, and then rescaling this range back to 0–100% reflectance. We then extracted the spectra with maximum total brightness (area under the curve) for each patch. The variability between individuals of each species was assessed using pairwise distances in tetrahedral color space. If the patch measurements for the two individuals were very different in terms of color (separated by >0.1 Euclidean distance in color space), we inspected the spectral measurements to identify possible inaccurate readings. Nine spectra were removed from the data set after this process, leaving a total of 213 spectra used for analysis.

## Calculation of color variables used in analysis

To compare color diversity and color properties of different structures, we focused on five variables: (1) the number of voxels occupied in avian color space, (2) mean color distance in avian color space, (3) color saturation, (4) stimulation of the avian double cones, and (5) peak reflectance. These variables describe color diversity (1–2), color purity (3), perceptual brightness (4), and objective brightness (5), respectively. Peak reflectance is simply the maximum reflectance from each spectrum. Perceptual brightness was modeled as the photon catch from a chicken double cone (*Gallus gallus*, built-in data in the *pavo* package; see details above), since current evidence suggests that the double cones mediate achromatic/brightness perception in birds (*Hart, 2001*; *Jones and Osorio, 2004*). Saturation and color diversity were based on modeling spectra in avian color space (*Stoddard and Prum, 2008*). This space represents all the colors a bird can theoretically perceive. Relative cone stimulation was calculated from photon cone catches using cone sensitivity functions in *pavo*. Bird species vary in their ultraviolet spectral sensitivity; some species have a VS (violet-sensitive) cone type that is maximally sensitive in the violet range while others have a UVS (ultraviolet-sensitive) cone type that is maximally sensitive in the ultraviolet range (*Hart, 2001*). Because we modeled plumage colors across many phylogenetic groups, we used the sensitivity curves in *pavo* for an 'average UVS' ($\lambda_{max}$=372 nm) and 'average VS' ($\lambda_{max}$=416 nm) type system. Since results in general were similar for a UVS- and VS-type system, we only include analyses based on a VS-type visual model (summary statistics for a UVS-type cone can be found in *Supplementary file 1e-f*), which is the ancestral condition in birds (*Ödeen and Håstad, 2003*).

Saturation in tetrahedral color space is simply the distance from the center of the tetrahedron (r vector, as defined by *Stoddard and Prum, 2008*). For number of voxels occupied, we followed the approach of *Delhey, 2015*. The tetrahedral color space is divided into 3D pixels (voxels), and then the number of voxels that have at least one data point are counted. The resolution of raster cells was set to 0.1, which gives a total of 236 voxels in tetrahedral color space. Mean color span is a measure of the spread of samples in color space and is calculated as the mean of pairwise Euclidean distances between all samples. This measure is more robust to sample size differences than voxel occupancy, which makes it better suited for comparing the plumage colors in our data set.

## Statistical analysis

To compare iridescent structures recorded in the feather iridescence database (thickness of melanin layers, diameter of interior hollowness, and number of layers), we applied simulation-based phylogenetic analyses of variance (ANOVA), as described by *Garland et al., 1993* using the R package *phytools* (*Revell, 2012*). Since this function assumes Brownian evolution of traits, we measured phylogenetic signal (Pagel's lambda) in the traits tested to confirm that this assumption was not violated. All traits tested recorded a high and significant lambda (*Supplementary file 1b*). To clarify relationships between groups, we also performed phylogenetic pairwise *t*-tests where necessary (using the R package *phytools* [*Revell, 2012*], *Supplementary file 1a*). Species that had more than one entry in the database (e.g., from multiple studies or multiple patches) were averaged before analysis. For comparison, we also included melanosome diameters from black feathers in some analyses. These data were taken from *Li et al., 2012*.

We performed a test for multimodality to assess whether solid rods show a binary distribution, following the method described by *Fisher and Marron, 2001*, which is implemented in the R package *modetest* (*Ameijeiras-Alonso et al., 2018*).

To explore how melanosome modifications affect color production, we fitted separate linear models with response variables saturation, avian double cone stimulation, and peak reflectance. Avian double cone stimulation and peak reflectance were log-transformed before inclusion into the models to achieve normally distributed residuals. We used binary predictors to describe the absence/presence of the three melanosome modifications: thin melanin layers (<190 nm), hollowness and platelet shape. We also added the interaction term {hollowness×platelet}, since the optical effect of hollow platelets is not expected to be simply the addition of hollowness and platelet shape. This is because hollow platelets lower the refractive index of melanosome layers by having relatively less melanin in each layer. This property only applies to melanosomes that have both modifications (hollow and platelet) simultaneously. Note that since we have included an interaction term, the variables hollow and platelet are only describing a situation where the interaction is zero (i.e., for hollow rods and solid platelets, respectively).

Spectral data derived from optical simulations were analyzed using multiple linear regressions with the variables described above (summary of results can be found in *Supplementary file 1g-i*). For plumage data, we also needed to account for phylogenetic relatedness, as well as individual variability in patch color (for each species we had measurements from two individuals). We did this using Bayesian linear mixed models, adding phylogenetic structure and patch as random factors in the model. The phylogeny used was the same as for earlier analyses but pruned to contain only the 80 species in our plumage measurements. We also added a fourth predictor: presence/absence of a photonic crystal (PC). This variable accounts for expected variation in color brightness and saturation that is explained by whether the structure has a single layer of melanosomes or several (in the optical model simulations, all structures had four layers). We used the R package *MCMCglmm* (*Hadfield, 2010*) to run our Bayesian model with Markov Chain Monte Carlo methods. We ran chains for each model for 50 million generations, with a sampling frequency of 500. The first 50,000 generations were discarded as burnin. We used the default priors for the fixed effects and set an inverse gamma distribution prior for the variance of residuals and random factors. We checked that the analysis had reached a stable phase by visually examining trace plots and checked that autocorrelation values between parameters was low (all <0.1). We also formally tested convergence of the chain using Heidelberg's and Welch's convergence diagnostics (all variables passed both tests). Summary of results for each model can be found in *Supplementary file 1j-l*.

## Acknowledgements

The authors would like to thank the American Museum of Natural History (NYC) for allowing them to use the bird skin collections for plumage spectral color measurements. In particular, the authors thank Paul R Sweet, collections manager for the ornithological collections, for his help. The authors thank Kaspar Delhey for generously sharing R code to reproduce his measure of color diversity (voxel occupancy). The authors also thank Heinz Durrer for generously allowing them to reprint his microscopy images of iridescent nanostructures in Figure 2. Funding in support of this work was provided by Princeton University (MCS), a Packard Fellowship for Science and Engineering (MCS), the Grainger Bioinformatics Center (CME), and the National Science Foundation (Award 2029538 to MCS and

Award 2112468 to CME). The authors would also like to thank Raphael Steiner, Jarome Ali, Merlijn Staps, and two anonymous reviewers for helpful discussion and feedback on an earlier version of this manuscript.

## Additional information

### Funding

| Funder | Grant reference number | Author |
| --- | --- | --- |
| Princeton University | Graduate Student Fellowship | Klara Katarina Nordén |
| Packard Fellowship for Science and Engineering | | Mary Caswell Stoddard |
| National Science Foundation | 2029538 | Mary Caswell Stoddard |
| National Science Foundation | 2112468 | Chad M Eliason |
| Princeton University | | Mary Caswell Stoddard |
| Grainger Bioinformatics Center | | Chad M Eliason |

The funders had no role in study design, data collection and interpretation, or the decision to submit the work for publication.

### Author contributions

Klara Katarina Nordén, Conceptualization, Formal analysis, Investigation, Methodology, Project administration, Software, Validation, Visualization, Writing - original draft, Writing – review and editing; Chad M Eliason, Conceptualization, Funding acquisition, Investigation, Methodology, Software, Supervision, Validation, Writing – review and editing; Mary Caswell Stoddard, Conceptualization, Funding acquisition, Investigation, Methodology, Resources, Supervision, Validation, Writing – review and editing

### Author ORCIDs
Klara Katarina Nordén ⓘ http://orcid.org/0000-0003-0810-5280
Chad M Eliason ⓘ http://orcid.org/0000-0002-8426-0373
Mary Caswell Stoddard ⓘ http://orcid.org/0000-0001-8264-3170

### Decision letter and Author response
Decision letter https://doi.org/10.7554/eLife.71179.sa1
Author response https://doi.org/10.7554/eLife.71179.sa2

## Additional files

### Supplementary files
• Supplementary file 1. Supplementary tables - details of statistical test results, supplementary color statistics and source links to images in *Figures 1–3*. (a) Result of phylogenetic pairwise t-test for difference in melanin layer thickness. P values corrected for multiple comparisons. (b) Phylogenetic signal for traits used in phylogenetic t-tests and ANOVA. (c) Summary statistics for brightness and saturation of optical model data, subdivided by melanosome type (used in linear models). (d) Summary statistics for brightness and saturation of plumage data, subdivided by melanosome type (used in Bayesian linear models). (e) Color diversity and saturation for optical model data using the UVS cone sensitivity function. (f) Color diversity and saturation for plumage data using the UVS cone sensitivity function. (g) Summary of results for linear model of saturation for optical model data. Model: Saturation (r.vec)~ hollow + thin+ platelet + hollow*platelet. Residual standard error: 0.072 on 4495 degrees of freedom. Multiple R-squared: 0.170, adjusted R-squared: 0.169. (h) Summary of results for linear model of brightness (double cone quantum catch) for optical model data. Model:

Log brightness (double cone quantum catch)~ hollow + thin+ platelet + hollow*platelet. Residual standard error: 0.502 on 4495 degrees of freedom. Multiple R-squared: 0.459, adjusted R-squared: 0.459. (i) Summary of results for linear model of brightness (peak reflectance) for model data. Model: Log brightness (peak reflectance)~ hollow + thin+ platelet + hollow*platelet. Residual standard error: 0.4629 on 4495 degrees of freedom. Multiple R-squared: 0.540, adjusted R-squared: 0.540. (j) Summary of results for Bayesian linear model of saturation for plumage data. Model: saturation ( r.vec)~ hollow + thin+ platelet + hollow*platelet+ PC. (k) Summary of results for Bayesian linear model of brightness (double cone quantum catch) for plumage data. Model: Log brightness (double cone quantum catch)~ hollow + thin+ platelet + hollow*platelet+ PC. (l) Summary of results for Bayesian linear model of brightness (peak reflectance) for plumage data. Model: Log brightness (peak reflectance)~ hollow + thin+ platelet + hollow*platelet+ PC (m) Image attribution for feather in *Figure 1*. (n) Image attribution for photographs in *Figure 2*. (o) Image attribution for silhouettes in *Figure 3*.

• Transparent reporting form

### Data availability

All data generated or analyzed during this study are included in the manuscript and supporting files. All datasets, supporting code and raw data to reproduce analyzes have been deposited with Dryad (https://doi.org/10.5061/dryad.4j0zpc8bq).

The following dataset was generated:

| Author(s) | Year | Dataset title | Dataset URL | Database and Identifier |
|---|---|---|---|---|
| Nordén KK, Eliason CM, Stoddard MC | 2021 | Data and code from: Evolution of brilliant iridescent feather nanostructures | https://doi.org/10.5061/dryad.4j0zpc8bq | Dryad Digital Repository, 10.5061/dryad.4j0zpc8bq |

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
