## [Editor Report]

Nordén et al., examine feather iridescent color diversity across bird species. Their findings show how key modifications in feather melanosomes, pivotal nanophotonic structures, underlie the brilliant colors of iridescent feathers, broadening feather color range approximately twofold. In a next step, the authors evaluate the function of feather melanosomes by performing optical modeling of nanostructure diversity, evaluating up to 4500 distinct nanostructure combinations, which are then contrasted with the observed (color) spectral data from 111 plumage regions across 80 (diverse) bird species. This meticulous integration of diverse methods across a comprehensive dataset will not only inform biologists studying structural color biodiversity, but it may also inspire engineers designing nanophotonic systems.

---

## [Decision Letter]

**Decision letter after peer review:**

Thank you for submitting your article "Evolution of brilliant iridescent feather nanostructures" for consideration by *eLife*. Your article has been reviewed by two peer reviewers, and the evaluation has been overseen by a Reviewing Editor and Christian Rutz as the Senior Editor. The reviewers have opted to remain anonymous.

The Reviewing Editor has consolidated the reviewers' feedback and drafted this decision letter to help you prepare a revised submission.

Essential revisions:

1) The specific predictions for the relative size of melanin layers in a two-material arrangement (keratin and melanin) is very helpful. Hence, we would like to see similar theoretical predictions for the three-material arrangements (air, keratin, and melanin) such as in the case of the hollow melanosome. By completing the analysis, this manuscript will fully substantiate its findings.

2) Further, is there any evidence that hollow melanosomes are able to achieve the same results as solid melanosomes with a lesser volume of melanin? That could be a potential explanation for why there is little observed difference between the occupied color spaces. Please address this issue in the manuscript.

3) Could some of the observed discrepancy between the modeling and measured colors of solid platelets at long wavelengths be the result of using a wavelength-independent refractive index for melanin? We agree that this is likely to have little impact on the broad-scale patterns you find (and wavelength dependence is difficult to implement in MEEP), but it is an explanation worth exploring, as such factors often modify major conclusions in an informative fashion given biodiversity.

4) The introduction provides a nice overview of what is known so far and seems to contain most of the relevant articles. Please consider Maia et al., (2013) as a potentially helpful reference for line 28, or briefly explain why it is not relevant: Maia et al., 2013, Key ornamental innovations facilitate diversification in an avian radiation. Proceedings of the National Academy of Sciences, USA 110, 10687-10692.

5) Figure 1: The color blue (in the hue selected) is not the best example of iridescence. Readers less familiar with the subject might assume that all blues (e.g., blue jay) are produced using this mechanism. Maybe purple would be a better example, as this is more uniquely associated with iridescence? Either way, please inform the general reader of this key issue.

6) Figure 2: Please note that the “Melanosome modification” symbols are not always clearly associated. The figure would benefit from centralizing the symbols, adding borders to the lower layer, or similar more effective graphic design solutions: please refine holistically.

7) Line 110: Please define in what they differ, because “minimally modified” and “closely resemble” imply there are minor differences, but what are they? In general, the reader needs some clarifications and help here to comprehend these factors.

8) Line 137-139: We find this section unclear, because the units are the same – therefore, it is not intuitive which numbers relate to thickness and which to reflected wavelength (e.g., the 300 nm and 700 nm values). Maybe also add “melanin layer thicknesses of at most” before 38 nm just to avoid confusion with reflected wavelength. In general, these sentences need revision, so they are easier to understand.

9) Line 137-139: A space is needed between numbers and units, e.g., “300 nm” (not “300nm” etc.). Please resolve throughout the manuscript across all text submitted.

10) Line 139-141: Consider communicating this before the expected melanin layer thicknesses, to improve clarity.

11) Line 159: The functions of the layer of melanosome are diverse, so is it necessary to delimit the keratin layer? Isn’t keratin also delimited in birds without melanin? It would be helpful if you could clarify this section for our broad readership, providing a fuller perspective.

12) Figure 4: Please distinguish between thick rods and black feather rods, using a design that helps the reader.

13) Line 243: This is confusing because it suggests that “38-206 nm” is within the bird’s visual range while to our knowledge it is not. This can be resolved easily throughout the manuscript (e.g., line 263 is clear). It would also help to at least once discuss this point more explicitly, e.g. when the issue first arises.

14) Line 520: Consider citing Beltran et al., (2021) for integrating the observation that speciation rates are positively correlated with the rate of plumage color evolution in hummingbirds. In case this isn’t essential for your article’s thesis, please clarify in your response letter why: Beltrán et al., 2021, Speciation rates are positively correlated with the rate of plumage color evolution in hummingbirds. Evolution 75, 1665-1680.

15) Line 583: Rubenstein et al., (2021) also explored the genetic regulation of iridescent structures: Rubenstein et al., 2021, Feather gene expression elucidates the developmental basis of plumage iridescence in African starlings. Journal of Heredity, esab014. This study could be worth discussing.

16) Figure 4: Can we conclude all melanosomes with diameters <190 nm are associated with iridescence? And, what about slightly larger melanosomes? This type of question is relevant for several researchers including paleontologists. For example, given a fossil melanosome with diameters 200-250 nm aligned in a thin melanin layer, could we conclude they are iridescent? Please provide some guidance for interested readers.

17) The validation dataset is probably too small (80 species) to compare observed patterns across species with the optical simulations, given there are two orders of magnitude more extant bird species and not all simulation outcomes were recovered with the reasonable, but limited, dataset across real feathers. It seems worth discussing this shortcoming more extensively in the manuscript, providing directions for more thorough future comparisons (especially since specimens exist of most feathers of most species across a couple of major museums). Broader sampling seems like an essential next step for future studies.

18) The question posed in the Introduction – "Why have bird species with brilliant iridescence evolved not one but four different melanosome types?" – has not been solved entirely. Please briefly discuss the remaining gaps to be filled by future research.

19) To better understand the functional morphology workspace limit in relation to biodiversity, the reader would benefit from learning at what point melanosomes are too thick to induce iridescent color. Please clarify this point.

20) Points 17-19 above will help future studies harness the present dataset and modeling outcomes to test and answer related evolutionary and genetic questions that are out of scope of the current manuscript.

---

## [Author Response]

Essential revisions:1) The specific predictions for the relative size of melanin layers in a two-material arrangement (keratin and melanin) is very helpful. Hence, we would like to see similar theoretical predictions for the three-material arrangements (air, keratin, and melanin) such as in the case of the hollow melanosome. By completing the analysis, this manuscript will fully substantiate its findings.

This is an important point. Unfortunately, providing theoretical predictions for three-material arrangements is very challenging. In the current manuscript, we give the theoretical predictions for a two-material reflector, which are well understood. We can fit a structure with hollow melanosomes into this framework by assuming that the air layer has the same optical thickness as the keratin layers. Then, the same reasoning applies as for the melanin layers. We have attempted to make this point more clearly in Section 1 of the Results (“Identifying key melanosome modifications”) (together with the text about melanin layer thickness) (lines 163-174):

“To estimate the expected thickness of hollow interiors (air pockets), we can extend the argument for expected thickness of the melanin layer. If air pockets conform to the expected size range, this would suggest that they are tuned together with melanin layers to produce brilliant iridescence. Analogous to describing melanin rods as a melanin layer (Figure 2F), we can think of air pockets as an air layer. Since the equations above define reflection for a structure with only two materials (of high and low refractive index, respectively), we must assume that air layers have the same optical thickness as the keratin layers. Thus, both the keratin and air layers can be described by a single term, since (tk×nk)=(ta×na), where a denotes air and na=1. In this situation, the air layer should have a thickness of <350 nm to produce first order interference in the bird-visible spectrum—and a thickness of 75-175 nm to meet the condition for maximal reflectance. Thus, the expected range is 75-350 nm.”

However, the more difficult question of how a three-component multilayer could optimally be arranged (i.e., when all three layers can have any thickness) has no known global solution. Thus, in the field of materials science, three-component multilayers are optimized from simulations of many different combinations, not unlike the optical simulations we have provided in this manuscript (see for example Gautier, Julien, et al., “Study of normal incidence of three-component multilayer mirrors in the range 20–40 nm.” Applied optics 44.3 (2005): 384-390). It would be productive to derive the optimal arrangement of a three-component multilayer, but this would require a detailed study beyond the scope of the current manuscript. We have added the following sentences (lines 175-177) in the text to acknowledge these points:

“However, we note that a one-dimensional photonic crystal with three materials could have varying optical thickness for all three types of layers (where (tk×nk)≠(ta×na)). The optimal configuration of such a system is much harder to derive, making it difficult to generate specific predictions for this case.”

2) Further, is there any evidence that hollow melanosomes are able to achieve the same results as solid melanosomes with a lesser volume of melanin? That could be a potential explanation for why there is little observed difference between the occupied color spaces. Please address this issue in the manuscript.

This also a very interesting question worthy of further study. We expect melanin costs to be roughly equal for different structures since melanin layers conform to an optimal thickness range, no matter the type of melanosome. This is supported in the empirical data by the fact that melanin layer thickness is overlapping between all four derived melanosome types (Figure 5). However, if we look at the equation for producing interference colors with a multilayer ((tmel×nmel)+(tk×nk)=λ2,), there is room to adjust the melanin layer with respect to the keratin layer and achieve similar colors (as long as both add up to λ2). So, there is a possibility that different melanosome types may produce similar results with different amounts of melanin.

We can get a rough estimate of melanin cost by comparing the cross-sectional area of thin solid rods compared to hollow rods for species in our database where this information is available. Such a comparison shows that thin solid rods are expected to use less melanin than a hollow rod (Author response image 1). Since a hollow rod essentially contains two melanin layers in one melanosome, the total cost of melanin for a structure producing a similar color (rather than comparing melanosome to melanosome) is likely similar.

**Author response image 1. sa2fig1:** 

Thus, although small differences in melanin cost are possible among the different melanosome types, it is unlikely that great discrepancies in melanin cost exist for an overall structure—since two thin solid rods are roughly equivalent to one hollow rod in terms of color production.The data collected in the feather iridescence database are not particularly suitable for comparing melanin cost in detail, since we did not collect 3D information on melanosomes (it is also rarely available). Moreover, many hollow forms contain a honeycomb-like structure of melanin inside the air pocket, which should be accounted for. Thus, we do not feel that we can include a detailed investigation in the current study, but encourage future such studies.

We added the following sentences to the discussion to cover these points (line 671-678):

“An interesting question is therefore whether different types of iridescent structures contain different amounts of melanin. Our results show that structures with derived melanosomes have converged on a shared range of melanin layer thicknesses (Figure 5), which suggests that differences in melanin content may not be large. However, melanin content has never been compared across different iridescent structures. This would be an exciting avenue for future research, especially since differences in melanin production may induce pleiotropic effects on other traits, such as immune function or behavior (Ducrest et al., 2008).”

3) Could some of the observed discrepancy between the modelling and measured colors of solid platelets at long wavelengths be the result of using a wavelength-independent refractive index for melanin? We agree that this is likely to have little impact on the broad-scale patterns you find (and wavelength dependence is difficult to implement in MEEP), but it is an explanation worth exploring, as such factors often modify major conclusions in an informative fashion given biodiversity.

This is a good question and should definitely be considered as a factor in understanding discrepancies between the modelled and plumage data. However, it is unlikely that a wavelength-dependent refractive index could explain the lack of red colors in the plumage data for solid platelets. Melanin is an absorbing material; it has a complex refractive index with both a real part (related to refraction) and an imaginary part (related to absorbance). Both the real part and the imaginary part is slightly higher for short wavelengths and decreases towards longer wavelengths. A higher refractive index in the short wavelengths will act to increase reflectance – broadening and increasing a peak – but simultaneously the higher absorption will tend to decrease the amplitude of the peak. The result is a slightly broader (but not taller) peak in short compared to longer wavelengths. Thus, the expected overall effect would be to make shorter wavelength colors slightly brighter but less saturated and vice versa. It is likely that the effect of this modelled in a bird visual space may be quite small – but of course one cannot be sure before it has been done. Nevertheless, such an effect could not cause the absence of red colors for solid platelets, and in general no pronounced differences in this direction are seen between the plumage and modelled data.

We have added the following sentences in line 788-795:

“In reality, these values vary over the light spectrum for most materials. Both n and k decrease from short wavelengths to longer wavelengths for melanin. A higher refractive index is expected to broaden and increase reflection peaks, while a high extinction coefficient will tend to decrease the amplitude of the reflectance peak. Thus, peaks in the short wavelengths will tend be slightly broader (but not taller), resulting in a brighter but less saturated color, compared to long wavelength colors. However, we did not observe any large differences between modelled and plumage data in this direction, and thus we expect the effects of a varying refractive index to be insignificant for the lager patterns we describe.”

4) The introduction provides a nice overview of what is known so far and seems to contain most of the relevant articles. Please consider Maia et al., (2013) as a potentially helpful reference for line 28, or briefly explain why it is not relevant: Maia et al., 2013, Key ornamental innovations facilitate diversification in an avian radiation. Proceedings of the National Academy of Sciences, USA 110, 10687-10692.

We have added Maia et al., 2013 to the references in this sentence.

5) Figure 1: The color blue (in the hue selected) is not the best example of iridescence. Readers less familiar with the subject might assume that all blues (e.g., blue jay) are produced using this mechanism. Maybe purple would be a better example, as this is more uniquely associated with iridescence? Either way, please inform the general reader of this key issue.

It is hard to pick a color which could not also be associated with non-iridescent feather structures, since blue/purple/green are all commonly produced by both iridescent barbule structures and non-iridescent barb structures. Thus, we don't feel that changing the color to purple would necessarily make this distinction clearer. Iridescent structural colors, as opposed to other structural colors, are defined in the introduction (line 28-29), and we note that the caption to Figure 1 specifies that the structure illustrated is iridescent (Iridescent plumage is produced by nanostructures in the feather barbules).

6) Figure 2: Please note that the "Melanosome modification" symbols are not always clearly associated. The figure would benefit from centralizing the symbols, adding borders to the lower layer, or similar more effective graphic design solutions: please refine holistically.

We have centralized the melanosome modification symbols in Figure 2 to make the connection to melanosome type clearer.

7) Line 110: Please define in what they differ, because "minimally modified" and "closely resemble" imply there are minor differences, but what are they? In general, the reader needs some clarifications and help here to comprehend these factors.

The differences between thick solid rods in iridescent and black feathers has not previously been quantified, thus no detailed description of their differences can be mentioned. Based on the descriptions and microscope images available of rods in iridescent and non-iridescent feathers, we assume that differences must be minor, since they closely resemble each other. However, we added "minimally modified" to make clear that smaller differences in for example diameter of the melanosomes may exist. We further investigate and discuss these minor differences between thick rods in iridescent and black feathers in the Results section Thin melanin layers (see also Figure 4). We have added the sentence below (line 112-114) to the paragraph to make this clear:

“For a more detailed analysis of thick solid rods in weakly iridescent versus black feathers, see the next section (“Evolution of modified melanosomes in iridescent structures”).”

8) Line 137-139: We find this section unclear, because the units are the same -- therefore, it is not intuitive which numbers relate to thickness and which to reflected wavelength (e.g., the 300 nm and 700 nm values). Maybe also add "melanin layer thicknesses of at most" before 38 nm just to avoid confusion with reflected wavelength. In general, these sentences need revision, so they are easier to understand.

We have rewritten the section (line 141-155) to make it clearer. We have also added a couple of sentences explaining how our calculated values relates to the expected range of 37.5-206 nm. The edited section is copied below:

“If we assume that the structure should reinforce wavelengths within the bird-visible spectrum (300-700 nm), we can calculate the range we should expect for melanin layer thickness, using 300 nm and 700 nm as endpoints. Here, we use the refractive indices nmel=2 for 300 nm and nmel=1.7 for 700 nm, following Stavenga et al., 2015. This gives us a maximum melanin layer thickness ranging from <75 nm (maximum thickness for reinforcing ultraviolet wavelengths) to <206 nm (maximum thickness for reinforcing red wavelengths), with maximum reflectance (where (tmel×rmel)=λ4) at layer thicknesses of 37.5 nm and 103 nm, respectively. Note that the maximum values of 206 and 75 nm represent a situation where the optical thickness of melanin layers alone equal λ2, and thus keratin layers must be zero. Such a structure would not function as a photonic crystal, since it consists of a single thick layer of melanin. Thus, for iridescent structures producing first order interference peaks, we expect melanin layer thickness to be below 206 nm. Moreover, we expect a lower limit at 37.5 nm, since melanin layer thickness is unlikely to have evolved below the thickness required for maximum reflectance at ultraviolet wavelengths. This gives us an expected range of 37.5-206 nm.”

9) Line 137-139: A space is needed between numbers and units, e.g., "300 nm" (not "300nm" etc.). Please resolve throughout the manuscript across all text submitted.

We have added spaces throughout the manuscript after units.

10) Line 139-141: Consider communicating this before the expected melanin layer thicknesses, to improve clarity.

We have moved the sentence describing refractive indices so that it appears before the calculation of melanin thicknesses.

11) Line 159: The functions of the layer of melanosome are diverse, so is it necessary to delimit the keratin layer? Isn't keratin also delimited in birds without melanin? It would be helpful if you could clarify this section for our broad readership, providing a fuller perspective.

All iridescent structures will tend to have a well-defined keratin cortex since the melanosomes are arranged in ordered layers. Since the melanosome layers are enclosed in keratin, the outer cortex “layer” will naturally form. The optical effects of this cortex layer will be greatest for structures with a single melanin layer, which is why the effect of the cortex is not a focus of our study. The paragraph in question explores why thick rods are only found in single-layered structures. We suggest that this is because the thickness of the melanin layer does not actually matter in these structures, as only the overlying keratin cortex functions optically as a thin-film (as supported by studies of such structures). Thus, in this section, we choose not to discuss the functions of keratin or cortices more generally, which (though interesting) is beyond the scope of our study.

12) Figure 4: Please distinguish between thick rods and black feather rods, using a design that helps the reader.

We have added a dotted line for the distinction between thick solid rods and thin solid rods (thick solid rods being defined as < 190nm). Distinction between thick solid rods in black feathers and iridescent feathers is shown with color (black and grey, respectively).

13) Line 243: This is confusing because it suggests that "38-206 nm" is within the bird's visual range while to our knowledge it is not. This can be resolved easily throughout the manuscript (e.g., line 263 is clear). It would also help to at least once discuss this point more explicitly, e.g. when the issue first arises.

Changed sentence to: We predicted that if thin melanin layers did evolve for an optical benefit, they should have converged on the expected range for producing bright interference peaks in the bird-visible spectrum, i.e. a layer thickness between 37.5-206 nm.

The reason that this is the predicted optimal thickness is explained in more detail in Results section 1.

14) Line 520: Consider citing Beltran et al., (2021) for integrating the observation that speciation rates are positively correlated with the rate of plumage color evolution in hummingbirds. In case this isn't essential for your article's thesis, please clarify in your response letter why: Beltrán et al., 2021, Speciation rates are positively correlated with the rate of plumage color evolution in hummingbirds. Evolution 75, 1665-1680.

Thanks for the suggestion of including this recent exciting study; we have added it in the text (line 617).

15) Line 583: Rubenstein et al., (2021) also explored the genetic regulation of iridescent structures: Rubenstein et al., 2021, Feather gene expression elucidates the developmental basis of plumage iridescence in African starlings. Journal of Heredity, esab014. This study could be worth discussing.

This study is indeed important in that it explores an area that is very little studied. We now cite it in line 721.

16) Figure 4: Can we conclude all melanosomes with diameters <190 nm are associated with iridescence? And, what about slightly larger melanosomes? This type of question is relevant for several researchers including paleontologists. For example, given a fossil melanosome with diameters 200-250 nm aligned in a thin melanin layer, could we conclude they are iridescent? Please provide some guidance for interested readers.

Yes, this is an important insight to take away from our study that is relevant for paleontological studies. It does seem to be safe to assume that melanosomes with diameters <190nm where almost certainly part of an iridescent structure – with the caveats that fossilised melanosomes may be diagenetically altered. More speculatively – they could have been part of a photonic crystal, giving rise to brilliant iridescence. Unfortunately, keratin is usually not preserved in fossil feathers, making it impossible to know the original 3D configuration of the iridescent structure. Both thin melanin layers and a photonic crystal organization is required for brilliant iridescence, so it is also possible that melanosomes with thin melanin layers were part of a single-layer structure, producing weaker iridescence. We also want to emphasize that this boundary (<190 nm) should not be interpreted as an instantaneous shift to brilliant iridescence, but rather as the starting point for a gradual shift in color saturation. We have made this clearer by including an additional supplemental figure (Figure 6—figure supplement 2), showing the change in saturation for simulated structures with rods of increasing diameter.

On the other hand, we can say for certain that the structure would not have produced brilliant iridescence if the melanin layer thickness is >190 nm. For example, the average diameter of melanosomes in the feathers of Microraptor, a theropod predicted to exhibit iridescence, was 196 nm. This suggests a weak iridescence.

We have added the following sentences to our discussion to include these points (line 570-589):

“For the production of weakly iridescent colors, it is sufficient to organize a single layer of melanosomes of any size, since it is typically the thickness of the overlying keratin cortex that controls the interference color (Doucet, 2006; Maia et al., 2009). In contrast, to produce brilliant iridescence, we show that two key optical innovations are required: a photonic crystal (multiple periodic layers of melanosomes) and melanin layers with an optical thickness <λ/2. Indeed, Durrer, 1977 observed that these two features were common to the brilliant structures he studied and here we validate the importance of his observation with optical modeling and plumage color measurements. Specifically, we find that saturation increase for structures with melanin layer thickness <190 nm (Figure 6A-E, Figure 6—figure supplement 2), which we define as "thin melanin layers". Above this value, iridescent structures produce colors that have low saturation and brightness, irrespective of the number of melanosome layers (Figure 6A). This insight could be used to place a lower bound on when brilliant iridescence first evolved in feathers, using the fossil record. For example, the preserved melanosomes from the plumage of Microraptor, a feathered theropod that is predicted to have exhibited iridescent plumage, have an average diameter of 196 nm (Li et al., 2012). This suggests that Microraptor exhibited weak iridescence, as opposed to brilliant iridescence. However, we caution that preserved melanosomes with melanin layers <190 nm do not necessarily prove that the feathers originally produced brilliant iridescence. Both a photonic crystal and thin melanin layers are required to produce brilliant iridescence—and the three-dimensional structure is usually lost in fossil feathers.”

17) The validation dataset is probably too small (80 species) to compare observed patterns across species with the optical simulations, given there are two orders of magnitude more extant bird species and not all simulation outcomes were recovered with the reasonable, but limited, dataset across real feathers. It seems worth discussing this shortcoming more extensively in the manuscript, providing directions for more thorough future comparisons (especially since specimens exist of most feathers of most species across a couple of major museums). Broader sampling seems like an essential next step for future studies.

Broad sampling is important, and we agree that a larger sample would give us increased detail and may close some gaps between optical modeling and plumage data. In particular, it will be important to pair a larger plumage color data set with detailed TEM analysis describing the structures. Few of the studies in the feather iridescence database included information on the number of melanosome layers in photonic crystals, which is likely an important parameter to account for differences in the real data. A perhaps more significant difference between the plumage measurements and the model simulations is the fact that the plumage measurements are taken from feathers with considerable micro- and macro-structure which can influence color, while the model simulations only account for the nanostructure. How different barbule shapes affect colors is an active field of research but has not yet been explored for iridescent structures. Such an investigation would be beyond the scope of the current study – but we believe would probably be a fruitful way to understand remaining differences between optical model simulations and plumage measurements. We have added these points to the discussion in line 642-656:

“Our plumage color dataset, though phylogenetically broad, is relatively small, and it is possible that increased sampling could reveal smaller differences in color production between derived melanosome types. It would be important to pair a larger dataset with detailed information on the number of layers in the iridescent structures, since this is a parameter for which we could not fully control (number of layers are rarely reported in the literature). Future studies should also investigate potential interactions with the feather micro- or macrostructure. Iridescent feathers are known to have highly modified barbules (Durrer, 1977), which has been shown to affect coloration in Lawes' Parotia (Stavenga et al., 2011) and the African Emerald Cuckoo (Chrysococcyx cupreus, Harvey et al., 2013). The interaction of feather microstructure and coloration is an active field of study (McCoy et al., 2021), but how nanostructures and microstructures may interact is an unexplored topic. Such an investigation would likely explain some of the discrepancies between the plumage data (which is measured from many feathers with micro- and macro-shape) and the optical model simulations (which considers only the nanostructure).”

18) The question posed in the Introduction – "Why have bird species with brilliant iridescence evolved not one but four different melanosome types?" – has not been solved entirely. Please briefly discuss the remaining gaps to be filled by future research.

We suggest that evolutionary history may be a main factor determining the diversity of melanosome types seen in different groups, and thus, studying the development and evolution of different structures should be a key focus of future studies. We suggest that this may be done by carefully studying the evolution of microstructures in a clade that has repeated transitions to brilliant iridescence, combined with genetic and developmental studies (line 718-721). We also discuss why some clades have very variable melanosome types (fast color evolution), and potential ways our hypothesis could be tested (633-638).

However, we agree that there are many further interesting questions to investigate. We have extended the discussion of gaps to be filled in line 647-678, discussing how feather micro- and macrostructure offer interesting further avenues of research. We also stress that iridescent color production from single-layered structures should be investigated in more detail and have added a Figure 6—figure supplement 1 to support this point. We have also added some sentences on the potential non-signaling functions of different iridescent structures, and the possibility that different costs of melanin production might influence the evolution of certain melanosome types.

19) To better understand the functional morphology workspace limit in relation to biodiversity, the reader would benefit from learning at what point melanosomes are too thick to induce iridescent color. Please clarify this point.

We describe in line 1891-197 how single-layered structures usually function as thin-films, where the thickness of the overlaying keratin determines the color. In such a system, it the thickness of the melanin layer does not matter much, since it simply delimits the keratin cortex. In such a structure, to ask how thick the melanin layer could be would be the analogue of asking: how thick can the substrate be to still create iridescence from an oil-slick? Any thickness, since it is really the thickness of the oil film that determines the color. Similarly, in single-layered structures, it is (generally) the thickness of the keratin cortex that determines the color. So, the simple answer to the question—when are melanosomes are too thick to induce any iridescent color?—is that there is no such limit. This is why, we suggest, thick rods found in single layered structures are very similar to melanosomes found in black feathers – there's simply no need to modify them by much.

We would argue that it is more interesting to know at what point melanosomes are too thick to produce colors that improve on this simpler design of a thin-film, i.e. that produce brilliant iridescence. In PCs, melanin layers are no longer simply the substrate, but rather part of the periodic structure that gives rise to interference. The answer to this question is >190 nm, which we show with our study. However, it is not the case that photonic crystals with melanosomes >190 do not produce any interference colors, rather the interference produced is so weak (being of higher orders) that it does not significantly improve on a color that could be produced with a single keratin film.

Therefore, there is no limit at which melanosomes cannot produce iridescent color. The thicker they become, they will only create peaks in the visible spectrum of higher orders, which will be very pale. Since a single-layered structure with a keratin cortex of the right thickness already produces equal colors to such a structure once melanosomes >190, there is no benefit derived from multilayered structures with such large melanosomes. This places an absolute constraint on melanosome size to achieve brilliant iridescence (from photonic crystals), but not weak iridescence (from thin-film structures).

We have clarified these points in the discussion, by adding the following sentences (line 570-581):

“For the production of weakly iridescent colors, it is sufficient to organize a single layer of melanosomes of any size, since it is typically the thickness of the overlying keratin cortex that controls the interference color (Doucet, 2006; Maia et al., 2009). In contrast, to produce brilliant iridescence, we show that two key optical innovations are required: a photonic crystal (multiple periodic layers of melanosomes) and melanin layers with an optical thickness <λ/2. Indeed, Durrer, 1977 observed that these two features were common to the brilliant structures he studied and here we validate the importance of his observation with optical modeling and plumage color measurements. Specifically, we find that saturation increase for structures with melanin layer thickness <190 nm (Figure 6A-E, Figure 6—figure supplement 2), which we define as "thin melanin layers". Above this value, iridescent structures produce colors that have low saturation and brightness, irrespective of the number of melanosome layers (Figure 6A).”

20) Points 17-19 above will help future studies harness the present dataset and modeling outcomes to test and answer related evolutionary and genetic questions that are out of scope of the current manuscript.

Thank you very much for this feedback.